# Single-PanIN-seq unveils that *ARID1A* deficiency promotes pancreatic tumorigenesis by attenuating *KRAS*-induced senescence

Shou Liu[1†], Wenjian Cao[1,2†], Yichi Niu[1,2†], Jiayi Luo[1,3], Yanhua Zhao[1], Zhiying Hu[1], Chenghang Zong[1,2,3,4,5]*

[1]Department of Molecular and Human Genetics, Baylor College of Medicine, Houston, United States; [2]Genetics and Genomics Graduate Program, Baylor College of Medicine, Houston, United States; [3]Cancer and Cell, Biology Graduate Program, Baylor College of Medicine, Houston, United States; [4]Dan L Duncan Comprehensive Cancer Center, Baylor College of Medicine, Houston, United States; [5]McNair Medical Institute, Baylor College of Medicine, Houston, United States

**ABSTRACT** ARID1A is one of the most frequently mutated epigenetic regulators in a wide spectrum of cancers. Recent studies have shown that *ARID1A* deficiency induces global changes in the epigenetic landscape of enhancers and promoters. These broad and complex effects make it challenging to identify the driving mechanisms of *ARID1A* deficiency in promoting cancer progression. Here, we identified the anti-senescence effect of *Arid1a* deficiency in the progression of pancreatic intraepithelial neoplasia (PanIN) by profiling the transcriptome of individual PanINs in a mouse model. In a human cell line model, we found that *ARID1A* deficiency upregulates the expression of aldehyde dehydrogenase 1 family member A1 (*ALDH1A1*), which plays an essential role in attenuating the senescence induced by oncogenic KRAS through scavenging reactive oxygen species. As a subunit of the SWI/SNF chromatin remodeling complex, our ATAC sequencing data showed that *ARID1A* deficiency increases the accessibility of the enhancer region of *ALDH1A1*. This study provides the first evidence that ARID1A deficiency promotes pancreatic tumorigenesis by attenuating *KRAS*-induced senescence through the upregulation of *ALDH1A1* expression.

*For correspondence:
czong@bcm.edu

†These authors contributed equally to this work

Competing interest: The authors declare that no competing interests exist.

## Introduction

Pancreatic ductal adenocarcinoma (PDAC) was the third leading cause of cancer-related death in the United States in 2018 and is projected to become the second leading cause of cancer-related death by 2030 (*Rahib et al., 2014*). Besides the clear driver mutations of PDAC, which include *CDKN2A*, *TP53, SMAD4*, and *KRAS*, there are a large number of genes with low-frequency mutations (*Waddell et al., 2015*; *Bailey et al., 2016*; *Cancer Genome Atlas Research Network, 2017*). The statistical significance of a large number of low-frequency mutations suggests that they likely play functional roles in the promotion of tumor development. From the evolutionary perspective of tumor development, the probability of acquiring one of the mutations in low-frequency-mutation genes is significantly higher than the acquisition of one of the mutations in the very few high-frequency-mutation genes such as *TP53*, especially at the very early stage when the number of pre-lesion cells is limited. Therefore, it is reasonable to believe that a portion of the low-frequency mutations could be acquired at the early stage and that these mutations could play essential roles in tumorigenesis.

Among the genes with low-frequency mutations in PDAC, the top hit is *ARID1A* (a subunit of the SWI/SNF chromatin remodeling complex) with an 8% mutation rate (*Figure 1—figure supplement 1*). In addition to PDAC, *ARID1A* is also frequently mutated in other cancer types, with 45.2% mutation rate in ovarian cancer, 18.7% in gastric cancer, 18.6% in bladder cancer, 13.7% in hepatocellular cancer, 11.5% in melanoma, 9.4% in colorectal cancer, 8.2% in lung cancer, and 2.5% in breast cancer (*Kadoch et al., 2013*). Therefore, it is crucial to study the ubiquitous mechanisms for *ARID1A*-deficiency-facilitated tumorigenesis in various types of cancers.

ARID1A is involved in the regulation of many biological processes of cells, including differentiation, proliferation, and apoptosis (*Wu and Roberts, 2013*). In pancreatic cancer, recent studies have shown that *ARID1A* is necessary to maintain terminal differentiation of pancreatic acinar cells, and knockout of *ARID1A* results in the accelerated formation of acinar-to-ductal metaplasia (ADM) and then pancreatic intraepithelial neoplasia (PanIN) lesions (*Zhang, 2018*; *Livshits et al., 2018*; *Wang et al., 2019*). Although *ARID1A* depletion can prime acinar cells for early-stage PanIN lesion formation by facilitating shifts in cell identity (*Livshits et al., 2018*), the underlying mechanisms for the acceleration of *ARID1A*-deficiency-promoted PanIN progression remain elusive. Besides PanIN lesions, *ARID1A* deficiency could also facilitate intraductal papillary mucinous neoplasm (IPMN) formation through multiple pathways, including MYC-mediated protein synthesis (*Wang et al., 2019*) and SOX9/mTOR pathway (*Kimura et al., 2018*).

To dissect the mechanisms whereby *Arid1a* knockout drives PanIN progression, we applied the single-cell RNA-seq method to profile the transcriptome of individual early-stage PanIN lesions from *Arid1a* knockout and wildtype mice. Our results showed that *Arid1a* knockout could effectively reduce KRAS-induced senescence in PanIN lesions. It is important to point out that cellular senescence has been shown to be the major rate-limiting step in KRAS-driven PanIN progression (*Morton et al., 2010*; *Serrano et al., 1997*; *Li et al., 2018*). Therefore, with significant attenuation of senescence, *Arid1a* knockout can achieve significant acceleration of PanIN progression. Mechanistically, we found that aldehyde dehydrogenases play an essential role in attenuating senescence by scavenging the reactive oxygen species (ROS) induced by oncogenic *KRAS*.

## Results

### Individual PanIN lesion RNA-Seq unveils the potential player contributing to the attenuation of *Kras*-induced senescence in *Arid1a* knockout mice

To identify the mediators that contribute to *Arid1a*-deficiency-promoted PanIN progression, we followed the PanIN progression in conditional *Arid1a* knockout mice with mutant *Kras* (*Arid1a*<sup>fl/fl</sup>;*Lox-Stop-Lox-Kras*<sup>G12D/+</sup>;*Ptf1a*<sup>CreERT/+</sup>, we abbreviate it as AKC) and mice without *Arid1a* knockout (*Arid1a*<sup>+/+</sup>;*Lox-Stop-Lox-Kras*<sup>G12D/+</sup>;*Ptf1a*<sup>CreERT/+</sup>, we abbreviate it as KC) (*Figure 1—figure supplement 2A*). Consistent with the findings from other groups (*Zhang, 2018*; *Livshits et al., 2018*; *Wang et al., 2019*), we observed that *Arid1a* knockout facilitated the progression of lesions from ADM to PanIN3 (*Figure 1—figure supplement 2B,C*). At as early as the 2-month time point, the percentages of ADM, PanIN-1, and PanIN-2 were 74, 26, and 0%, respectively, in KC mice versus 53, 47, and 0% in AKC mice. At the 6-month time point, the percentages of ADM, PanIN-1, PanIN-2, and PanIN-3 were 77, 23, 0, and 0%, respectively, in KC mice versus 16, 74, 9, and 0.5% in AKC mice (*Figure 1—figure supplement 2C*). We observed the decrease in the percentage of ADM and increase in the percentage of high-grade PanIN lesions, indicating the accelerated progression of lesions.

To profile the transcriptome of individual PanIN lesions, we combined laser capture microdissection (LCM) with a highly sensitive single-cell RNA-seq method (MATQ-seq) developed in our lab (*Sheng et al., 2017*; *Figure 1A*). In total, we dissected and profiled 20 lesions from two KC mice and 24 lesions from two AKC mice. We only dissected PanIN-1 and PanIN-2 lesions because their duct-like structures can be easily recognized for dissection from frozen sections (*Figure 1—figure supplement 3A*). We observed that the lesions dissected from KC and AKC mice were clearly separated in the multidimensional scaling (MDS) plot, and 861 differentially expressed genes (DEGs) (*Supplementary file 2*), including 532 genes upregulated and 329 genes downregulated in AKC, were identified, which suggests the successful transcriptome profiling of individual PanIN lesions by MATQ-seq (*Figure 1B*).

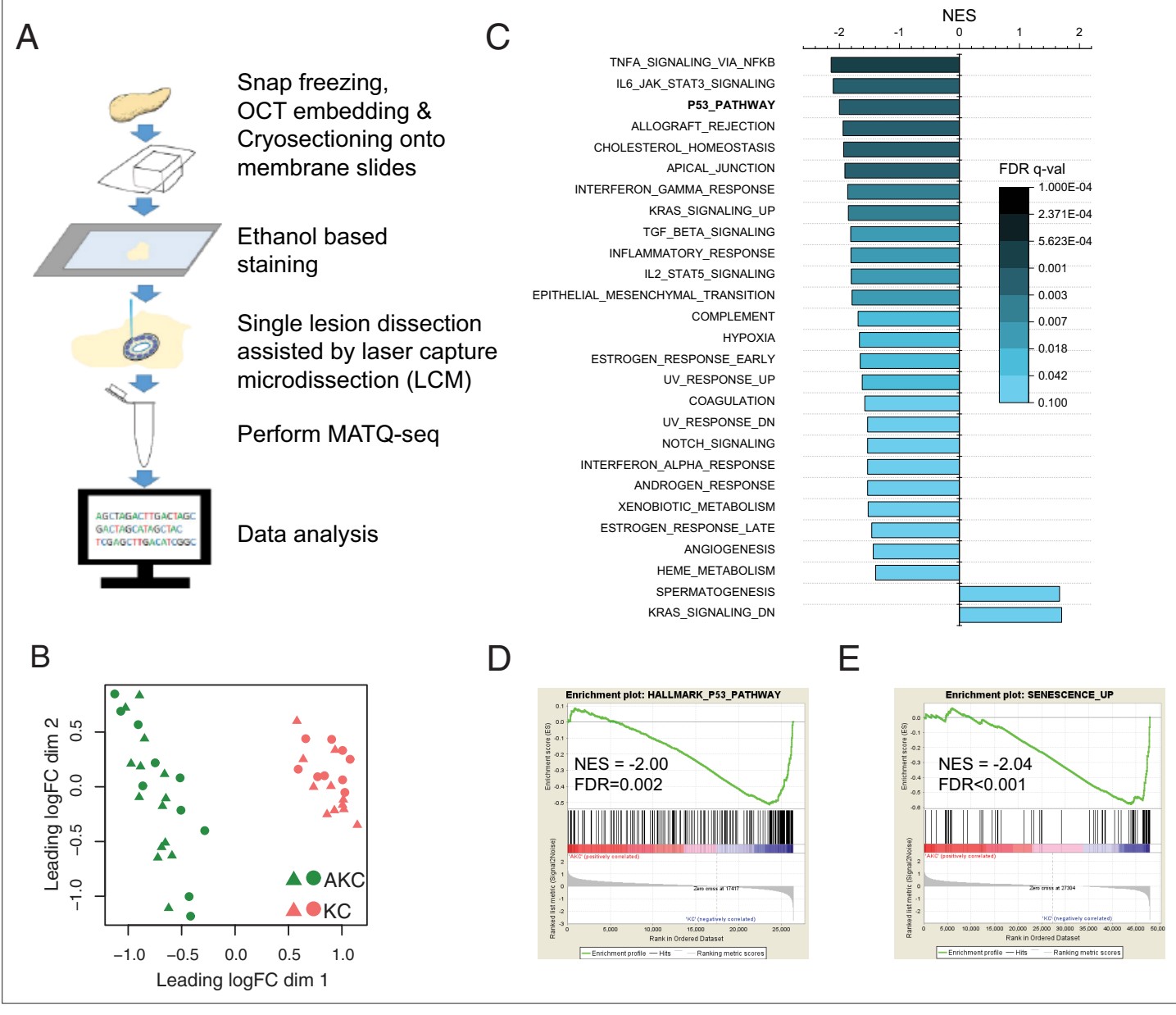

**Figure 1.** Single pancreatic intraepithelial neoplasia (PanIN) lesion RNA-seq unveils the potential player contributing to the attenuation of *Kras*-induced senescence in *Arid1a* knockout mice. (**A**) Experimental scheme of transcriptome profiling of single PanIN lesions. (**B**) Multidimensional scaling plot showed a clear separation of the transcriptome profiles of lesions from AKC mice (*Arid1a*fl/fl;*LSL-Kras*G12D/+;*Ptf1a*CreERT/+) and KC mice (*Arid1a*+/+;*LSL-Kras*G12D/+;*Ptf1a*CreERT/+). 24 lesions from 2 AKC mice and 20 lesions from 2 KC mice were used for single-lesion RNA sequencing. (**C**) Hallmark gene sets that are significantly enriched between lesions from AKC and KC mice. (**D, E**) Enrichment plots of P53_PATHWAY and SENESCENCE_UP from gene set enrichment analysis.

The online version of this article includes the following figure supplement(s) for figure 1:

**Figure supplement 1.** Mutations in *ARID1A* in ICGC-PACA-AU and TCGA pancreatic ductal adenocarcinoma (PDAC) cohorts based on cBioPortal data portal.

**Figure supplement 2.** *Arid1a* knockout drastically accelerates pancreatic intraepithelial neoplasia (PanIN) progression.

**Figure supplement 3.** Profile of the transcriptomes of individual pancreatic intraepithelial neoplasia (PanIN) by PanIN-seq.

**Figure supplement 4.** The gene set enrichment plots of 27 relevant pathways with statistical significance in Afl/flKC lesions.

With DEGs observed between AKC and KC lesions, we performed gene set enrichment analysis (GSEA) using Hallmark gene sets (*Liberzon et al., 2015*) to interrogate the pathways perturbed by *Arid1a* knockout. We observed that 25 gene sets were downregulated, and surprisingly, only two gene sets were upregulated in AKC lesions (*Figure 1C*, *Figure 1—figure supplement 4*, and *Supplementary file 1*, false discovery rate (FDR) < 0.1). It is worth noting that among the 27 gene sets 2 are specifically associated with *Kras* activation: KRAS_SIGNALING_UP (the gene set upregulated upon *Kras* activation) and KRAS_SIGNALING_DN (the gene set downregulated upon *Kras* activation). The gene set KRAS_SIGNALING_UP was downregulated while the gene set KRAS_SIGNALING_DN was upregulated (*Figure 1C* and *Figure 1—figure supplement 4*). This observation suggests that the activities of *Kras* signaling are partially impaired by *Arid1a* deficiency.

Furthermore, we observed that the *Tp53* signaling pathway was suppressed in AKC lesions (*Figure 1D*). It has been well established that upregulation of the *Tp53*-related pathway is closely associated with apoptosis or senescence. *ARID1A* mutations have also been shown to be mutually exclusive with *TP53* mutations in endometrial cancer (*Wu et al., 2017*). To determine whether *Arid1a* is involved in the regulation of apoptosis, senescence, or both, we further examined the activity of related pathways in *Arid1a* KO lesions. Interestingly, we found that the senescence-associated signaling pathway is significantly suppressed in lesions from AKC mice (*Figure 1E*). In contrast, the pathway activity associated with apoptosis was not significantly changed (*Figure 1—figure supplement 3B*). These observations led us to hypothesize that *Arid1a* deficiency could promote PanIN lesion progression through the attenuation of *Kras*-induced senescence.

Furthermore, senescent cells feature senescence-associated secretory phenotype (SASP), including high levels of inflammatory cytokines and immune modulators. With the attenuation of senescence promoted by *Arid1a* deficiency, we expected to observe reduced inflammatory response in the GSEA. Indeed, we observed that multiple signaling pathways associated with inflammation, including TNFα signaling, IL6–STAT3 signaling, IL2–STAT5 signaling, IFN-α signaling, and IFN-γ signaling, were significantly suppressed in *Arid1a* KO lesions (*Figure 1C*, *Figure 1—figure supplement 4*, and *Supplementary file 1*).

## In vivo, ex vivo, and in vitro verification of the attenuation of *Kras*-induced senescence by *Arid1a* deficiency

To verify the effect of *Arid1a* deficiency on *Kras*-induced senescence, we performed senescence-associated beta-galactosidase (SA-β-Gal) staining on lesions from KC and AKC mice. SA-β-Gal-positive lesions were observed in five out of seven (71%) KC mice. In contrast, only one out of six (17%) AKC mice showed SA-β-Gal-positive lesions. Among the mice with SA-β-Gal-positive lesions, the percentage of SA-β-Gal-positive lesions in KC mice was about twice of that in AKC mice (*Figure 2A,B*). These data confirmed that *Arid1a* knockout indeed reduced *Kras*-induced senescence.

To further verify the effects of *Arid1a* knockout on senescence, we performed an ex vivo culture experiment using acinar cells isolated from AKC and KC mice. SA-β-Gal staining was performed to examine the senescence of acinar cells. As shown in *Figure 2C,D*, due to flat morphology of senescent cells and the large cell-size variation, we cannot accurately quantify the number of SA-β-Gal-negative cells. Instead of using the percentage of senescence cells, here we quantified the senescence based on the intensity of SA-β-Gal staining of the positive cells. We observed that the intensity of SA-β-Gal staining in acinar cells from AKC mice was significantly less than that from KC mice. Importantly, this result suggests that the effect of *Arid1a* knockout on senescence is likely an intrinsic cellular response rather than a microenvironmental response such as accelerated clearance of senescent cells by immune cells (*Pignolo et al., 2020*; *Baker et al., 2011*).

Based on the results of the ex vivo experiment, we next tested whether we could observe similar effects of *ARID1A* deficiency on *KRAS*-induced senescence in in vitro cell lines. We established a human pancreatic Nestin-expressing (HPNE) cell line (an intermediary cell type formed during ADM) with inducible KRAS$^{G12D}$ expression (iKRAS-HPNE cells). The induction of KRAS activity was verified by western blot of both KRAS and phosphorylated ERK (*Figure 2—figure supplement 1A*). Next, we knocked out *ARID1A* by CRISPR-Cas9 in iKRAS-HPNE cells. Two isogenic HPNE clones were used in the following experiments (*Figure 2—figure supplement 1B*), and the knockout efficiency of *ARID1A* was further confirmed by qRT-PCR (*Figure 2—figure supplement 1C*).

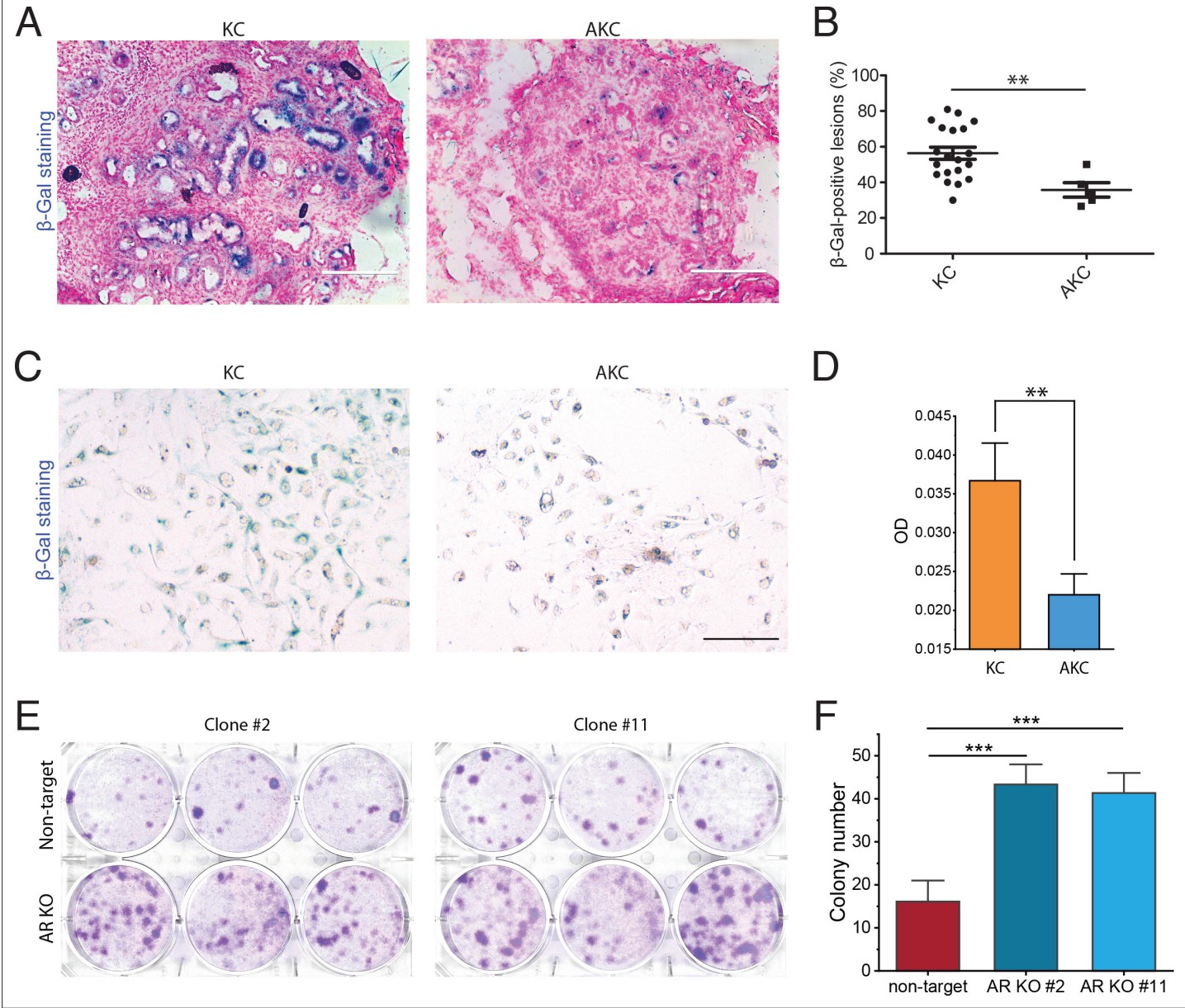

**Figure 2.** In vivo, ex vivo, and in vitro verification of attenuation of *Kras*-induced senescence by *Arid1a* deficiency. (**A**) Representative images of senescence-associated beta-galactosidase (SA-β-Gal) staining of frozen pancreatic sections from KC mice and AKC mice. (**B**) SA-β-Gal-positive lesions were counted at five random fields under the microscope in four KC mice and one AKC mouse, presented as percentages. (**C**) Representative images of SA-β-Gal staining of ex vivo culture from KC and AKC mice 1 month after administration of tamoxifen. (**D**) Quantification of the intensity of SA-β-Gal staining at five random fields under the microscope. (**E**) Colony formation assay of *ARID1A* knockout cells and wildtype human pancreatic Nestin-expressing (HPNE) cells with KRAS induction by doxycycline (6 μg/ml) for 15 days. (**F**) Quantification of colony number in panel (**E**). The colony formation assay was performed twice. Student's t-test: **p<0.01; ***p<0.001. Scale bars: 200 μm.

The online version of this article includes the following figure supplement(s) for figure 2:

**Figure supplement 1.** Generation of cell line with inducible *KRAS* overexpression and *ARID1A* knockout.

To examine the effect of *ARID1A* deficiency on *KRAS*-induced senescence, we first performed SA-β-Gal staining in *ARID1A* knockout (*ARID1A*-KO) and wildtype iKRAS-HPNE cells upon KRAS induction. We observed that *ARID1A*-KO cells showed a slightly lower percentage of SA-β-Gal-positive cells compared with wildtype iKRAS-HPNE cells (*Figure 2—figure supplement 1D*). The in vitro result of SA-β-Gal staining is less significant than what we observed both in vivo and ex vivo. One possible reason for this discrepancy is that in terms of evaluation of senescence SA-β-Gal staining may

not be as effective in immortalized cell lines as in tissue samples (*Dimri et al., 1995*). Colony formation assay is an alternative method to measure senescence. Here, we performed a colony formation assay to evaluate the ability of cells to escape from *KRAS*-induced senescence. We observed that knockout of *ARID1A* drastically increased the number of cells escaping from *KRAS*-induced senescence (*Figure 2E,F*).

## ARID1A knockout reduces KRAS activities and inflammatory response

With the consistent in vivo, ex vivo, and in vitro observation of the anti-senescence effects of *ARID1A* deficiency, we next investigated the molecular mechanisms by which *ARID1A* knockout promotes the escape from *KRAS*-induced senescence. We first performed RNA-seq on *ARID1A*-KO (clone #2) and wildtype HPNE cells with or without *KRAS* induction. We observed a clear separation between *ARID1A*-KO and wildtype HPNE cells under both conditions in the MDS plot (*Figure 3A*).

Next, we performed GSEA using the Hallmark gene sets. Interestingly, we found the gene set KRAS_SIGNALING_UP was downregulated in *ARID1A*-KO cells compared with wildtype cells (*Figure 3—figure supplement 1A,B*), which is consistent with our data from PanIN lesions. To verify this observation, we examined the activity of ERK, a classical downstream effector of KRAS signaling, in *ARID1A*-KO and wildtype cells. As shown in *Figure 3—figure supplement 2*, the phosphorylation of ERK upon KRAS induction in *ARID1A*-KO cells was significantly reduced in comparison with the wildtype cells. The consistent observation between in vivo and in vitro indicates that *ARID1A* deficiency partially impairs the activities of KRAS signaling.

Next, we performed an interaction test on the gene expression data to identify the genes that have different responses to *KRAS* activation depending on *ARID1A* status (*Supplementary file 4*). GSEA results showed that six signaling pathways were remarkably repressed in *ARID1A*-KO HPNE cells, and four of them are involved in inflammatory response (*Figure 3—figure supplement 3A,B*), which is consistent with what we observed in PanIN lesions from AKC mice (*Figure 1C*).

Furthermore, we examined the expression of a classic marker of cellular senescence: *CDKN1A*. We observed that the activation of *CDKN1A* expression upon *KRAS* induction was also significantly reduced in the *ARID1A*-KO cell line (*Figure 3—figure supplement 3C,D*), which indicates that our HPNE cell line model successfully recapitulated the senescence phenotypes observed in the mouse model.

## ARID1A knockout significantly upregulates the expression of aldehyde dehydrogenase (ALDH) family members

To identify the underlying players that promote the attenuation of cellular senescence, we analyzed the DEGs between wildtype and the *ARID1A*-KO cell line with or without *KRAS* induction (*Figure 3B*, *Figure 3—figure supplement 4A*, and *Supplementary file 3*). For the first clone with *KRAS* induction, we identified 125 upregulated genes and 165 downregulated genes between the wildtype and the *ARID1A*-KO line (*Figure 3B*). To exclude the genes whose expression changes may be associated with mutant KRAS signaling, we compared the list of DEGs under two conditions: with or without *KRAS* induction. As shown in *Figure 3C*, for the upregulated genes, 57 out of 125 genes (46%) are shared between the two conditions and the expected number of random overlap is 1.79. For the downregulated genes, 54 out of 165 genes (33%) are shared and the expected number of random overlap is 3.38. These results indicate that these genes are mainly dependent on *ARID1A* deficiency.

Among the DEGs between *ARID1A*-KO and wildtype HPNE cells, *ALDH1A1* exhibits the significant change in differential gene expression for both conditions: with or without *KRAS* induction (*Figure 3B and D*, *Figure 3—figure supplement 4A*, and *Supplementary file 3*). To rule out any potential clonal bias, we also performed RNA-seq on a second clone (clone #11). We observed that *ALDH1A1* was also significantly upregulated in the second clone under both conditions (*Figure 3—figure supplement 4B–D* and *Supplementary file 3*). The upregulation of *ALDH1A1* in *ARID1A*-KO cells was further verified by both qRT-PCR (*Figure 3—figure supplement 4E*) and western blot (*Figure 3E*). Considering that *ALDH1A1* has been shown to participate in the clearance of ROS (*Raha et al., 2014*) and ROS are vital mediators of *KRAS*-induced senescence (*Storz, 2017*), we hypothesize that *ALDH1A1* is the gene that mediates the effect of *ARID1A* deficiency on *KRAS*-induced senescence.

Next, we examined our PanIN-seq data to evaluate the expression of *Aldh1a1* and other members of the *ALDH* family. Interestingly, we observed that *Aldh3a1* is significantly

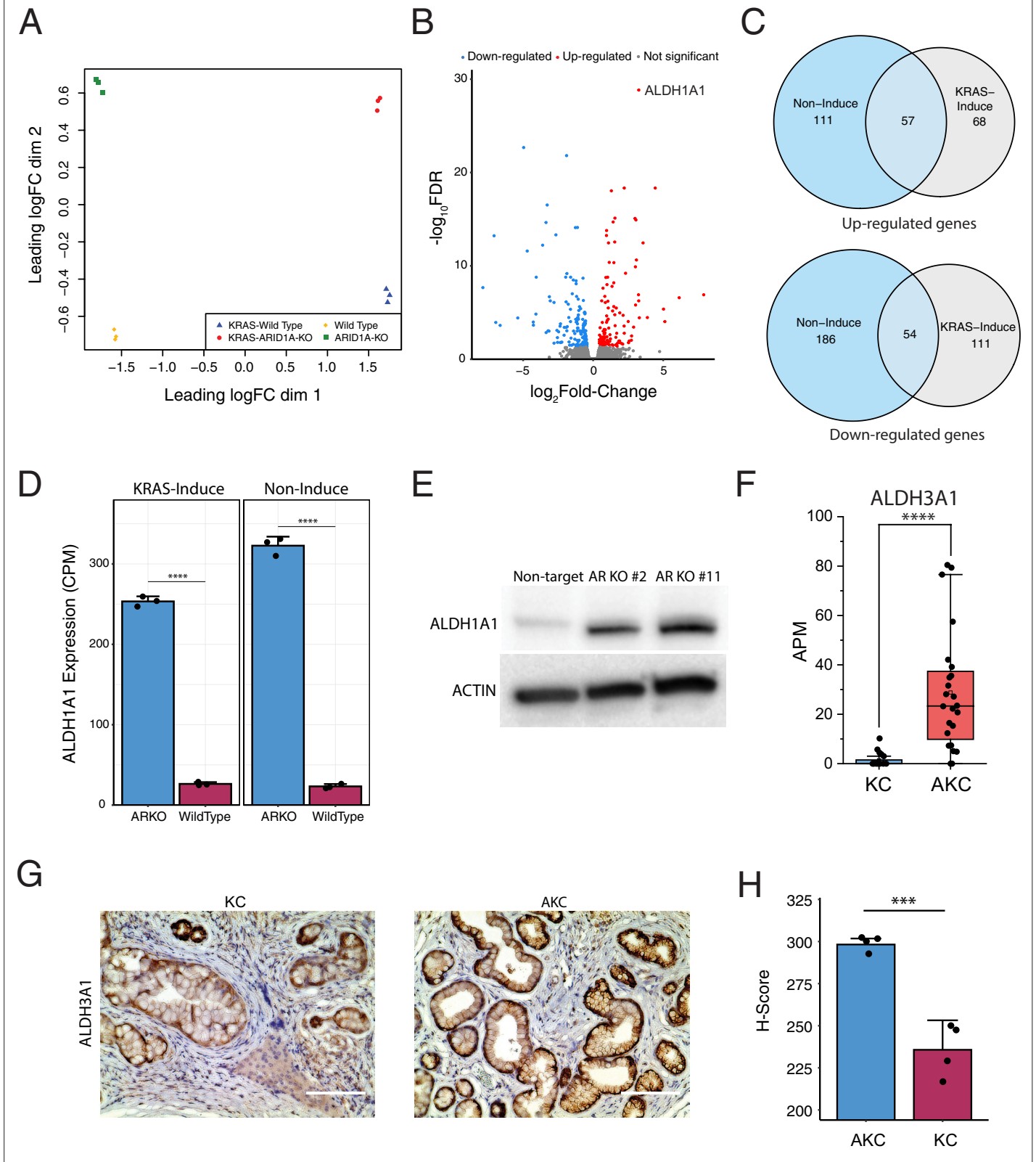

**Figure 3.** *ARID1A* knockout upregulates aldehyde dehydrogenase (ALDH) expression. (**A**) Multidimensional scaling plot demonstrated clear separation between the transcriptome profiles of *ARID1A*-KO human pancreatic Nestin-expressing (HPNE) cells and wildtype cells with or without KRAS induction. RNA sequencing was performed with three biological repeats. (**B**) Volcano plot of differentially expressed genes between *ARID1A* knockout cells and wildtype cells with KRAS induction. (**C**) Venn diagram showing the upregulated genes (upper) and downregulated genes (bottom) that are shared

*Figure 3 continued on next page*

*Figure 3 continued*

between cells with (gray) or without (blue) KRAS induction. (**D**) *ALDH1A1* mRNA levels quantified by sequencing data are significantly different between *ARID1A*-KO cells and wildtype cells with (left) or without (right) Kras induction. CPM: count per million reads. (**E**) Western blot for ALDH1A1 expression in *ARID1A*-KO cells and wildtype cells with KRAS induction. (**F**) mRNA level of *Aldh3a1* in KC and AKC lesions based on pancreatic intraepithelial neoplasia (PanIN)-seq data. APM: amplicon per million reads. (**G**) IHC staining against ALDH3A1 in KC and AKC lesions. Scale bars: 200 µm. (**H**) Comparison of ALDH3A1 levels between KC and AKC lesions based on the intensity of staining in (**G**). H-score was calculated by counting the number of lesions with different levels of staining intensity at four random fields under the microscope. Student's t-test: ***p<0.001; ****p<0.0001.

The online version of this article includes the following figure supplement(s) for figure 3:

**Figure supplement 1.** Gene set enrichment analysis on RNA-seq data.

**Figure supplement 2.** *ARID1A* knockout impairs phosphorylation of ERK in human pancreatic Nestin-expressing (HPNE) cells upon KRAS induction.

**Figure supplement 3.** Differential response to oncogenic KRAS in *ARID1A*-KO and wildtype cells.

**Figure supplement 4.** *ALDH1A1* expression in *ARID1A* knockout human pancreatic Nestin-expressing (HPNE) cells.

upregulated in lesions from AKC mice (*Figure 3F* and *Supplementary file 2*), which was further confirmed by immunohistochemistry (IHC) staining (*Figure 3G,H*). This result suggests that in different species different types of *ALDH* family proteins can be used to mediate the attenuation of *Kras*-induced senescence in *Arid1a*-deficient cells.

## *ARID1A* KO facilitates escape from *KRAS*-induced senescence via *ALDH1A1*

Given the important role of ALDH in ROS clearance, a high level of ALDH could also be important for the development of *KRAS*-driven PDAC. Here, we analyzed the expression of ALDH family members in normal pancreas and PDAC samples (*Bailey et al., 2016*; *GTEx Consortium, 2013*). In normal pancreas tissues, we mainly observed the expression of *ALDH1A1* (*Figure 4—figure supplement 1A*), with different cell types exhibiting different expression levels of *ALDH1A1* (*Figure 4—figure supplement 1B*). Since the tumor cells are mainly epithelial cells, we only compared PDAC data to pancreatic ductal cells to avoid the confounding factors caused by the cell type difference. As shown in *Figure 4—figure supplement 1B*, there are four subclusters of normal ductal cells. The average expression level of *ALDH1A1* in normal pancreatic ductal cells (clusters 1–3) is less than 50. We excluded cluster 4 since the *ALDH1A1*-positive cells are indicative of the ductal stem cell population (*Rovira et al., 2010*). In contrast to the expression levels in normal ductal cells, we observed that in 63% of PDAC samples, the expression levels of *ALDH1A1* are higher than 50 TPM, and in 10% of samples, the expression levels are higher than 200 TPM (*Figure 4* and *Figure 4—figure supplement 1C*). Furthermore, we examined the mutation levels in *ALDH1A1*. We observed that only 0.2% of the patients (1 out of 576 patient samples from two cohorts [*Bailey et al., 2016*; *Cancer Genome Atlas Research Network, 2017*]) acquired mutations in *ALDH1A1* (*Figure 4B*). This observation further supports our hypothesis that *ALDH1A1* plays an important role in *KRAS*-driven PDAC development.

Next, to validate the essential role of *ALDH1A1* in promoting the escape of cells from *KRAS*-induced senescence, we performed a colony formation assay in HPNE cells with and without N,N-diethylaminobenzaldehyde (DEAB, a pan-inhibitor of ALDH) treatment. We observed that inhibition of ALDH1A1 activity significantly decreased the number of colonies formed in *ARID1A* knockout cells; in contrast, no significant changes were observed in the wildtype cells (*Figure 4C,D*). To rule out the unknown effects of DEAB on HPNE cells, we also performed a colony formation assay on *ARID1A*-KO HPNE cells with and without *ALDH1A1* knockdown. The knockdown efficiency was verified by qRT-PCR (*Figure 4—figure supplement 2*). We also observed that the colony number in *ARID1A*-KO cells with *ALDH1A1* knockdown was significantly less than that without *ALDH1A1* knockdown (*Figure 4E,F*), which is consistent with the results of the ALDH inhibitor experiment.

Furthermore, we examined the levels of ROS production in *ARID1A*-KO cells and wildtype cells. We observed that the fraction of ROS-positive cells in *ARID1A*-KO iKRAS-HPNE cells was significantly less than in wildtype cells, regardless of KRAS induction (*Figure 4G*). To verify the role of *ALDH1A1* in reducing ROS, we measured the ROS level in *ARID1A*-KO cells with *ALDH1A1* knockdown. We observed that the ROS level in *ALDH1A1* knockdown cells was significantly higher than that observed in the cells with scramble shRNA (*Figure 4H*), indicating that *ALDH1A1* is responsible for the clearance of ROS in HPNE cells with *ARID1A* knockout. From the results above, we conclude that *ARID1A*

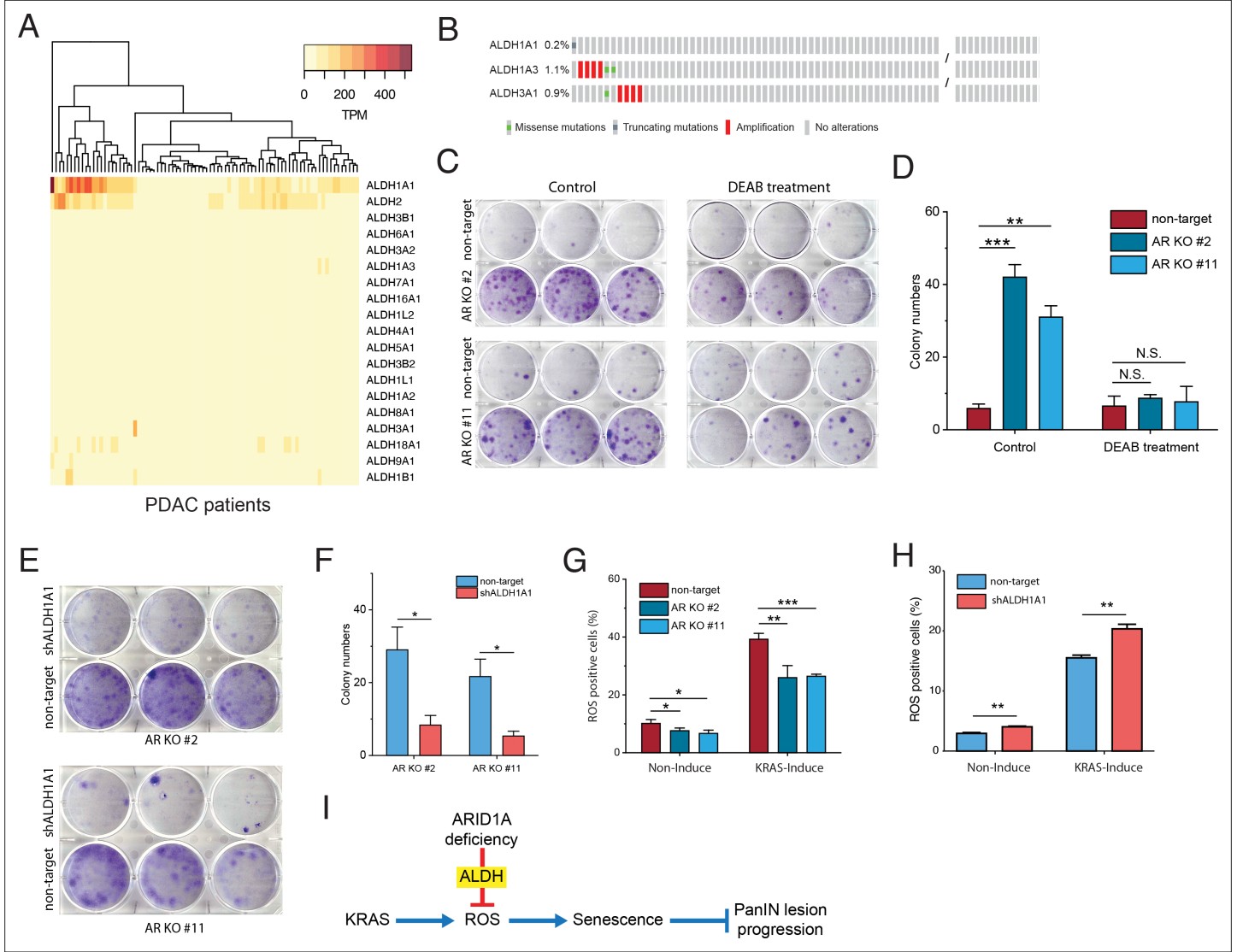

**Figure 4.** *ARID1A* knockout facilitates escape from *KRAS*-induced senescence by upregulating *ALDH1A1* expression. (**A**) Heatmap of the expression levels of aldehyde dehydrogenase (ALDH) family members in pancreatic ductal adenocarcinoma (PDAC) patients (*Bailey et al., 2016*). (**B**) Mutation rates of *ALDH1A1*, *ALDH1A3*, and *ALDH3A1* in PDAC patients. The mutation data from these two studies (*Bailey et al., 2016*) and TCGA (*Cancer Genome Atlas Research Network, 2017*) were used for analysis. (**C**) Colony formation assay of *ARID1A* knockout cells and wildtype cells with KRAS induction, treated with and without ALDH inhibitor DEAB. (**D**) Quantification of colony number in panel (**C**). The colony formation assay was performed twice. (**E**) Colony formation assay of *ARID1A* knockout cells expressing shRNA targeting *ALDH1A1* and scramble shRNA control. (**F**) Quantification of colony number in panel (**E**). The colony formation assay was performed twice. (**G**) Measurement of the reactive oxygen species (ROS) level using an H2DCFDA-based ROS detection assay kit. Percentage of positive cells measured by flow cytometry. (**H**) Measurement of the ROS levels in the *ARID1A*-KO cells with *ALDH1A1* knockdown or scramble shRNA. (**I**) Working model for *ARID1A*-deficiency-promoted pancreatic intraepithelial neoplasia (PanIN) lesion progression via inhibition of ROS production. Student's t-test: *p<0.05; **p<0.01; ***p<0.001.

The online version of this article includes the following figure supplement(s) for figure 4:

**Figure supplement 1.** The expression of aldehyde dehydrogenase (ALDH) family members in normal pancreas and pancreatic ductal adenocarcinoma (PDAC).

**Figure supplement 2.** Knockdown efficiency of *ALDH1A1* in human pancreatic Nestin-expressing (HPNE) cells with *ARID1A* knockout (clone #2 and #11) was confirmed by qRT-PCR.

knockout can effectively reduce cellular ROS levels by upregulating the expression of *ALDH1A1*, which then leads to significant attenuation of *KRAS*-induced senescence and acceleration of PanIN progression (*Figure 4I*).

## *ARID1A* KO alters gene expression mainly by modulating the chromatin accessibility of distal regulatory elements

As one of the DNA binding subunits of the SWI/SNF complex, ARID1A can regulate gene expression by modulating the chromatin accessibility for transcription factor (TF) binding, recruitment of coactivators/corepressors, and facilitation of chromatin looping required to approximate promoters with distal enhancers (*Wu and Roberts, 2013*). To investigate how *ARID1A* deficiency enhances the expression of *ALDH1A1*, we performed ATAC-seq on *ARID1A*-KO and wildtype HPNE cells. The distribution of the fragment sizes and the distance from transcription start sites for both wildtype and *ARID1A*-KO are shown in *Figure 5—figure supplement 1A–F* (*Supplementary file 5*). As shown in *Figure 5A*, the Spearman correlation coefficients between knockout and wildtype cells are significantly lower than those between replicate samples in both groups (*Figure 5B* and *Figure 5—figure supplement 1G*), suggesting that the changes in DNA accessibility were robustly captured.

Next, we examined the accessibility of the promoter and enhancer regions of the genes that are differentially expressed between *ARID1A*-KO cells and wildtype cells. We separated the DEGs into two groups based on the fold change. The genes with positive fold-change values are the genes upregulated in *ARID1A*-KO cells (upregulated genes group in *Figure 5B*), and the genes with negative fold-change values are the genes downregulated in *ARID1A*-KO cells (downregulated genes group in *Figure 5B*). We plotted a scatter plot of read counts in peaks between wildtype and *ARID1A*-KO for promoters (*Figure 5C*) and enhancers (*Figure 5D*) and observed that the number of peaks affected by *ARID1A* deficiency in the distal regulatory regions is significantly larger than the number of peaks in promoter regions.

The heatmap of reads for the differential peaks is shown in *Figure 5E*, and the average read density profiles are shown in *Figure 5—figure supplement 2A,B*. For the differential peaks of promoters, the heatmaps of reads and the average read density profiles are shown in *Figure 5—figure supplement 2C,D*. We also performed analysis for the distribution of the differential peaks (*Figure 5—figure supplement 3A,B*) and a functional enrichment analysis (*Figure 5—figure supplement 3C,D*) using the GREAT algorithm (*McLean et al., 2010*). We observed consistent enrichment in enhancer regions. Significant interactions between the SWI/SNF complex and distal regulatory regions have also been observed in colon cancer (*Mathur et al., 2017*).

Next, we examined the association between DEGs and the peaks with differential accessibility. We observed that the number of DEGs is associated with peak changes with statistical significance for both promoters and enhancers (*Figure 5—figure supplement 4A–D*). We also noticed that the number of DEGs associated with peak changes in enhancer elements is significantly larger than the number of genes associated with peak changes in the promoter regions. This result further supports that *ARID1A* knockout alters gene expression mainly by modulating the chromatin accessibility of the enhancer elements.

Considering the observation that *AR1D1A* deficiency impairs the activities of KRAS signaling pathways based on the GSEA of transcriptome data (KRAS_SIGNALING_UP in *Figure 3—figure supplement 1A*), next we examined the chromatin accessibility of the genes involved in KRAS signaling. We observed that in comparison to the wildtype cells, chromatin accessibility decreased in *ARID1A*-KO cells (*Figure 5—figure supplement 5*). This observation suggests that *AR1D1A* deficiency impairs the activities of the KRAS signaling pathways by partially impairing the chromatin accessibility of the genes downstream of the KRAS pathways.

We next performed motif enrichment analysis for the differential ATAC peaks. We separated the ATAC peaks with significant changes between *ARID1A*-KO and wildtype cells into four groups: distal peaks with increased accessibility in *ARID1A*-KO cells, distal peaks with decreased accessibility in *ARID1A*-KO cells, promoter peaks with increased accessibility in *ARID1A*-KO cells, and promoter peaks with decreased accessibility in *ARID1A*-KO cells. We then performed motif enrichment analysis by using the AME algorithm (*McLeay and Bailey, 2010*) on each group of ATAC peaks. Interestingly, we found that the binding motifs of *SRY*, *FOX* family, *CDX1*, *SOX5*, etc., were enriched in the ATAC peaks with significantly increased accessibility in *ARID1A*-KO cells (*Figure 5F*, *Figure 5—figure supplement 6A,B* and *Supplementary file 7*). The binding motifs of the *FOS-JUN* family, *NFE2*, *NF2L1*, etc., were significantly enriched in the ATAC peaks with significantly decreased accessibility (*Figure 5G*, *Figure 5—figure supplement 6B*, and *Supplementary file 7*), which is consistent with the results of a recent study showing ARID1A as a co-factor of AP-1 (*Sen et al., 2019*).

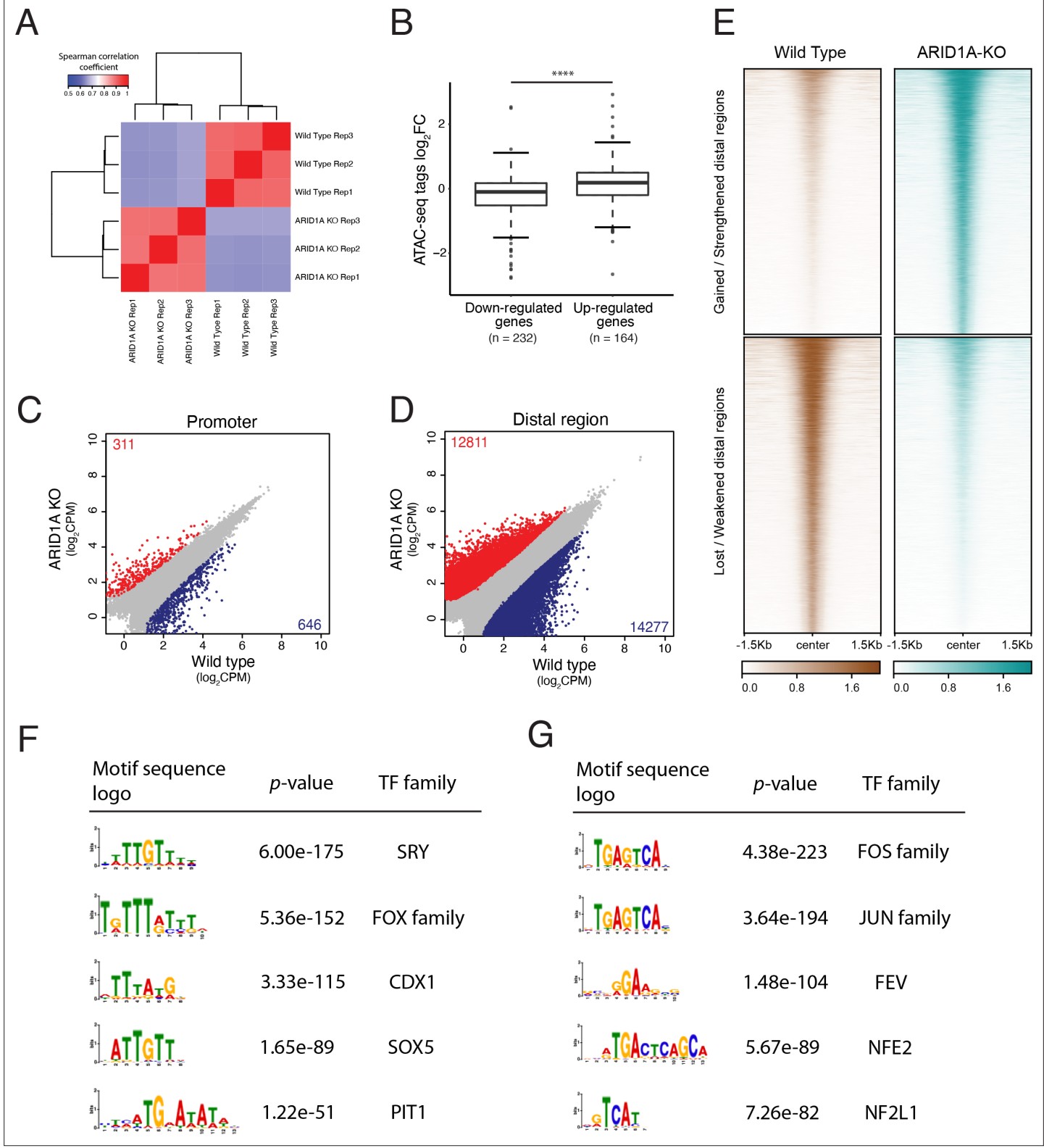

**Figure 5.** *ARID1A* knockout activates transcription of the ALDH1A1 gene by increasing the accessibility of its enhancer region. (**A**) Spearman correlation coefficients of the read counts in peaks between *ARID1A*-KO human pancreatic Nestin-expressing (HPNE) cells and wildtype cells. ATAC sequencing was performed with three biological repeats. (**B**) The average fold change of chromatin accessibility in the promoters and enhancers of differentially expressed genes identified in RNA-seq. Mann–Whitney–Wilcoxon test: ****p<0.0001. (**C, D**) The scatter plot of the read counts in each peak between ARID1A-KO cells and wildtype cells for promoter (**C**) and distal regions (**D**). The peaks with significantly increased read density in ARID1A-KO cells

*Figure 5 continued on next page*

*Figure 5 continued*

compared with wildtype cells are colored in red. The peaks with significantly decreased read density are colored in blue. (**E**) Heatmap of the gained or lost distal regions between *ARID1A*-KO cells and wildtype cells. (**F**) The top five transcription factor (TF) binding motifs significantly enriched in the enhancer ATAC peaks with increased accessibility in *ARID1A*-KO cells. (**G**) The top five TF binding motifs significantly enriched in the enhancer ATAC peaks with decreased accessibility in *ARID1A*-KO cells.

The online version of this article includes the following figure supplement(s) for figure 5:

**Figure supplement 1.** Quality control of ATAC-seq.

**Figure supplement 2.** Read density profiles of differential peaks.

**Figure supplement 3.** General analysis of ATAC-seq by GREAT.

**Figure supplement 4.** The overlap between the differentially expressed genes and differential peaks at promoter regions (**A, B**) and enhancer regions (**C, D**).

**Figure supplement 5.** Comparison in the change of chromatin accessibility between the genes within KRAS signaling pathway and the randomly chosen genes.

**Figure supplement 6.** Motif analysis on the promoter regions.

### *ARID1A* KO activates transcription of the *ALDH1A1* gene by increasing the accessibility of its enhancer region

To understand how the gene expression of *ALDH1A1* is altered by *ARID1A* knockout, we examined the accessibility of the promoter and distal regulatory elements for the *ALDH1A1* gene in both *ARID1A*-KO and wildtype cells. Interestingly, we found that there was a significant increase in accessibility in 9 out of 11 peaks at the distal regions when we compared *ARID1A*-KO cells with wildtype cells (*Supplementary file 6*). To further verify these functional regions, we compared the landscape of two well-known enhancer markers (H3K27ac and H3K4me1) in seven highly expressed *ALDH1A1* cell lines (ALDH1A1$^{high}$ cell lines) (*Mei et al., 2017*) with our ATAC-seq peaks (*Supplementary file 8*). We observed that five enhancer loci clearly overlapped with the active enhancer markers (*Figure 6A* and *Figure 6—figure supplement 1A,B*).

We also plotted out the landscape of H3K27ac and H3K4me1 in five lowly expressed *ALDH1A1* cell lines (ALDH1A1$^{low}$ cell lines) (*Mei et al., 2017*), and we did not observe any significant overlap between our ATAC-seq peaks and H3K27ac/H3K4me1-enriched regions (*Figure 6A*). For the five enhancer loci, we observed a consistent increase in accessibility in *ARID1A* knockout cells in comparison with wildtype cells (*Figure 6B*). Overall, these results confirm that *ARID1A* deficiency upregulates *ALDH1A1* expression by increasing the accessibility of the associated enhancer elements.

To identify proteins that could potentially bind to these five enhancer loci, we examined the TF ChIP-seq datasets from seven ALDH1A1$^{high}$ cell lines and five ALDH1A1$^{low}$ cell lines (*Mei et al., 2017*). We counted the binding events of each TF in the five candidate enhancer peaks for both ALDH1A1$^{high}$ cell lines and ALDH1A1$^{low}$ cell lines (*Supplementary file 9*). We list the TFs that could bind to the enhancer regions in *Figure 6C*. The TFs whose binding events were preferentially detected in the datasets from ALDH1A1$^{high}$ cell lines (Fisher's test, p<0.05) are indicated in red and include *EP300* and *NR3C1*. For the TFs that do not have enough datasets for a statistical test, we indicated the TFs whose binding events are observed in more than 50% of ALDH1A1$^{high}$ cell line datasets in blue, and the TFs whose binding events are observed in more than 25% but less than 50% of ALDH1A1$^{high}$ cell line datasets in black (*Figure 6C*, left panel). The TFs that are not expressed in HPNE cell lines were removed.

To verify the regulation of *ALDH1A1* expression by the TFs identified above, we knocked down *EP300* and *NR3C1*, respectively, in *ARID1A*-KO HPNE cell line (*Figure 6—figure supplement 2A,B*). In *Figure 6C* (middle and right panels), we observed that both knockdowns could significantly impair the transcription of the *ALDH1A1* gene. Collectively, these data also indicate that *ARID1A* deficiency promotes active transcription of *ALDH1A1* by altering genome accessibility and therefore allowing the binding of EP300 and/or NR3C1 to the corresponding enhancer loci.

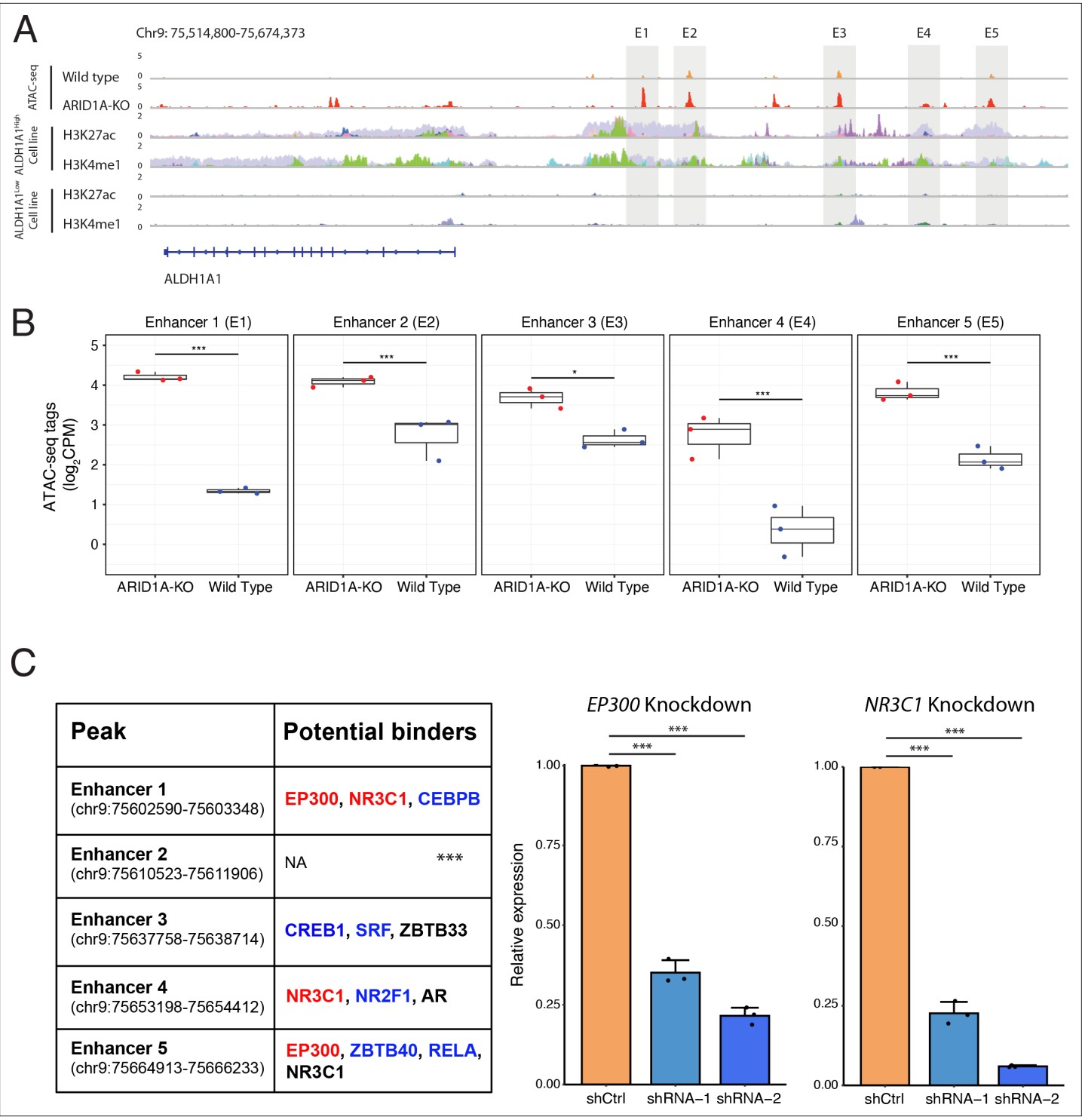

**Figure 6.** *ARID1A* knockout activates transcription of the *ALDH1A1* gene by increasing the accessibility of its enhancer region. (**A**) The ATAC-seq tracks and H3K4me1/H3K27ac ChIP-seq tracks of the distal regions of the *ALDH1A1* gene. The ChIP-seq tracks from different cell lines are labeled in different colors and overlaid. The figure with separated tracks is shown in *Figure 6—figure supplement 1*. The ALDH1A1high cell lines include A549, LOUCY, A673, 22RV1, VCAP, K562, and HepG2. The ALDH1A1low cell lines include HCT116, MCF7, Panc1, PC9, and DOHH2. The ChIP-seq data were obtained from the ENCODE database (*ENCODE Project Consortium, 2012*). (**B**) Read counts in the five enhancer peaks of the *ALDH1A1* gene in *ARID1A*-KO human pancreatic Nestin-expressing (HPNE) cells and wildtype cells. p-value: *p<0.05; ***p<0.001; ****p<0.0001. (**C**) Transcription factors (TFs) that regulate *ALDH1A1* expression by binding to its enhancer loci. Left panel: the list of TFs that can potentially bind to the enhancer regions of ALDH1A1 based on the ChIP-seq data from seven ALDH1A1high cell lines and five ALDH1A1Low cell lines. The TFs whose binding events are preferentially detected

*Figure 6 continued on next page*

*Figure 6 continued*

in the datasets from ALDH1A1[high] cell lines (Fisher's test, p<0.05) are marked in red. The TFs that are observed in more than 50% of the datasets from ALDH1A1[high] cell lines but do not have enough datasets for statistical tests are marked in blue. The TFs whose binding events are observed in more than 25% but less than 50% of the datasets from ALDH1A1[high] cell lines are marked in black. The TFs that are not expressed in HPNE cell lines were removed. Middle panel: expression of *ALDH1A1* in *EP300* knockdown cells and cells with scramble shRNA. Right panel: expression of *ALDH1A1* in *NR3C1* knockdown cells and cells with scramble shRNA.

The online version of this article includes the following figure supplement(s) for figure 6:

**Figure supplement 1.** The landscape of H3K27ac and H3K4me1 at the upstream of *ALDH1A1* gene in ALDH1A1[high] cell lines.

**Figure supplement 2.** Verification of knockdown efficiency.

## Discussion

As a *bona fide* oncogenic driver of pancreatic cancer, the mechanism by which mutant KRAS counteracts the oncogenic stress that it induces is critically important for the progression of precursor lesions (*Storz, 2017*). It has been shown that KRAS can upregulate the expression of multiple oxidoreductases via NRF2 (nuclear factor, erythroid derived 2, like 2) to counteract oncogenic stress (*DeNicola et al., 2011*). Meanwhile, KRAS can also enhance NADPH production by reprogramming the metabolism of glutamine to assist oxidoreductases in scavenging ROS (*Son et al., 2013*). However, the observation that a high percentage of senescent PanIN lesions occur in the pancreas of KC mice suggests that the mechanisms described above likely have limited effects in reducing *Kras*-induced ROS in neoplastic lesions. In the early stage of PDAC development, it is plausible that new mutations can be acquired to help cells to reduce ROS and escape the oncogenic *Kras*-induced senescence.

In this study, we successfully adapted a high-sensitivity total-RNA-based single-cell RNA-seq method, MATQ-seq, to profile the transcriptome of single lesions. We showed that transcriptome profiling of individual lesions is not only technically feasible but also it can provide important insights about the potential mechanisms of tumor progression. The transcriptome profiling of early lesions directly led us to unveil the effects of *ARID1A* knockout in attenuating *KRAS*-induced senescence and identify the important roles of *ALDH1A1* in mitigating the ROS stress induced by oncogenic KRAS.

It is worth noting that *ARID1A* has also been linked to the regulation of ROS through other pathways. Sun et al. found that *ARID1A* overexpression causes an increase in ROS by activating transcription of cytochrome P450 enzymes (CYP450) at the initiation stage of liver cancer (*Sun et al., 2017*). Interestingly, Ogiwara et al. found that *ARID1A* deficiency results in elevated ROS by inhibiting the transcription of *SLC7A11* (a transporter gene required for the import of cystine and the production of glutathione) in ovarian cancer (*Ogiwara et al., 2019*). Therefore, the effects of *ARID1A* deficiency on ROS are likely tissue-specific.

Besides reducing ROS levels through the upregulation of ALDH family proteins, we also observed that *ARID1A* deficiency directly suppresses the activities of KRAS signaling. Our in vitro and in vivo data showed that the activation of Hallmark gene sets KRAS_SIGNALING_UP and KRAS_SIGNALING_DN is remarkably impaired upon *ARID1A* knockout. Consistently, the activation of several inflammation-related signaling pathways upon Kras induction is inhibited in PanIN lesions and HPNE cells with *ARID1A* deficiency compared with wildtype counterparts, including TNFα signaling, IL6–STAT3 signaling, IL2–STAT5 signaling, IFN-α signaling, and IFN-γ signaling. Both reduction of KRAS activities and inflammatory cytokine response are also consistent with our observation of the increased attenuation of Kras-induced senescence upon ARID1A knockout.

Overall, experimental evidence shows that mutant KRAS signaling needs to be tuned to efficiently drive tumorigenesis: too much signaling could lead to cell growth arrest, while too little signaling could limit proliferation (*Li et al., 2018*). To find the 'sweet spot' of *KRAS* activation, *KRAS* could harness its own activities by changing the mutation pattern (including mutation position and type of substitution) and seeking cooperation with other genes (e.g., mutagenesis of tumor suppressors) through the course of passive selection during tumor development (*Li et al., 2018*). Overall, our study demonstrates the important roles of epigenetic mutations in harnessing KRAS activities and promoting tumorigenesis. Future studies of such effects in other frequently mutated epigenetic genes are greatly desired.

# Materials and methods

**Key resources table**

| Reagent type (species) or resource | Designation | Source or reference | Identifiers | Additional information |
|---|---|---|---|---|
| Genetic reagent (*Mus musculus*) | *Ptf1a*[CreERT] | Jackson Lab | No: 019378 | |
| Genetic reagent (*M. musculus*) | *Lox-Stop-Lox-Kras*[G12D] | Jackson Lab | No: 008179 | |
| Genetic reagent (*M. musculus*) | Arid1aflox | PMID:18448678 | | Dr. Zhong Wang lab (University of Michigan) |
| Cell line (*Homo sapiens*) | hTERT-HPNE | Dr. Jennifer Baily lab (UT Health) | RRID:CVCL_C466 | |
| Recombinant DNA reagent | pInducer20-KRAS[G12D] | Dr. Haoqiang Ying lab (MD Anderson Cancer Center) | | Inducible expression of KRAS[G12D] |
| Recombinant DNA reagent | pL-CRISPR.EFS.tRFP | Addgene | RRID:Addgene_57819 | Pol III-based sgRNA expression backbone |
| Recombinant DNA reagent | pGIPZ | Open Biosystems | | Pol III-based shRNA expression backbone |
| Sequence-based reagent | sgARID1A | This paper | sgRNA | CAGCGGTACCCGATGACCAT |
| Sequence-based reagent | shALDH1A1 | This paper | shRNA | GGAGTGTTTACCAAAGACATT |
| Sequence-based reagent | shEP300-1 | This paper | shRNA | CGGCAAACAGTTGTGCACA |
| Sequence-based reagent | shEP300-2 | This paper | shRNA | AGCTACTGAAGATAGATTA |
| Sequence-based reagent | shNR3C1-1 | This paper | shRNA | CCAACGGTGGCAATGTGAA |
| Sequence-based reagent | shNR3C1-2 | This paper | shRNA | AGCTGTAAAGTTTTCTTCA |
| Commercial assay or kit | ROS detection assay kit | BioVision | Cat# K936-250 | |
| Commercial assay or kit | Senescence β-Galactosidase Staining Kit | Cell Signaling Technology | Cat# 9860 | |
| Chemical compound, drug | ALDEFLUOR diethylaminobenzaldehyde (DEAB) reagent, 1.5 mM in 95% ethanol | Stemcell Technologies Inc | Cat# 01705 | |
| Antibody | (Rabbit polyclonal) anti-ALDH1A1 | Abcam | Abcam Cat# ab23375, RRID:AB_2224009 | WB(1:400) |
| Antibody | (Rabbit polyclonal) anti-Aldh3a1 | Abcam | Abcam Cat# ab76976, RRID:AB_1523110 | IHC(1:100) |
| Antibody | (Rabbit polyclonal) anti-Ras | Abcam | Abcam Cat# ab180772, RRID:AB_2884935 | WB(1:500) |
| Antibody | (Mouse monoclonal) anti-β-Actin | Sigma-Aldrich | Sigma-Aldrich Cat# A1978, RRID:AB_476692 | WB(1:4000) |
| Antibody | (Rabbit monoclonal) anti-Phospho-p44/42 MAPK (Erk1/2) | Cell Signaling Technology | Cell Signaling Technology Cat# 4370, RRID:AB_2315112 | WB(1:1000) |

## Mice

All animal experiments in this study were performed in accordance with a protocol approved by IACUC of Baylor College of Medicine. In this study, the following mice strains were generated: $Arid1a^{fl/fl}$; $Lox\text{-}Stop\text{-}Lox\text{-}Kras^{G12D/+}$;$Ptf1a^{CreERT/+}$, $Arid1a^{fl/+}$;$Lox\text{-}Stop\text{-}Lox\text{-}Kras^{G12D/+}$;$Ptf1a^{CreERT/+}$, and $Arid1a^{+/+}$;$Lox\text{-}Stop\text{-}Lox\text{-}Kras^{G12D/+}$;$Ptf1a^{CreERT/+}$. The mouse with the target allele type of $Lox\text{-}Stop\text{-}Lox\text{-}Kras^{G12D/+}$ was ordered from Jackson Laboratory (Stock No: 008179 | LSL-K-ras G12D). The promoter of $Ptf1a$ is used to drive the expression of inducible Cre in adult acinar cells (Stock No: 019378 | $Ptf1a^{CreERT/+}$, denoted as $C$). Removal of the floxed exon 8 of the $Arid1a$ (denoted as $A^{fl}$) leads to loss of $Arid1a$ transcript (Gao, 2008). Oncogenic $Kras$ mutant allele is silenced by a floxed STOP cassette ($Lox\text{-}Stop\text{-}Lox\text{-}Kras^{G12D/+}$, denoted as $K$). The adult mice (6–8 weeks) of the above genotypes were administered with tamoxifen (75 mg/kg/day for five consecutive days) to induce efficient ablation of $Arid1a$ and activation of oncogenic $Kras$ in pancreatic acinar cells.

## Cell lines

We received the HPNE cell line (RRID:CVCL_C466) from Dr. Jennifer Bailey's lab (UT Health Center), and the line is cultured in media recommended by ATCC. The identity of the cell line has been authenticated by STR profiling, and the mycoplasma contamination in cell lines has been routinely tested. HPNE with inducible $KRAS^{G12D/+}$ (iKRAS-HPNE) was generated by transduction of plasmid pInducer20-$KRAS^{G12D}$ into parental HPNE cells. After G418 selection for 15 days, the survived cells were then used for single-cell expansion in 96-well plate. For $ARID1A$ knockout HPNE cells, isogenic clone of iKRAS-HPNE was infected with lentivirus packaged with pL-CRISPR.EFS.tRFP-sgARID1A. After infection for 5 days, the RFP-positive cells were sorted and single-cell expansion was performed. $ARID1A$ knockout was confirmed by Sanger sequencing of the isogenic clones. The guide RNA sequence used for $ARID1A$ knockout is CAGCGGTACCCGATGACCAT. For $ALDH1A1$ knockdown cells, two $ARID1A$ knockout HPNE clones were infected by lentivirus packaged with plasmid pGIPz-$ALDH1A1$. After infection for 5 days, the GFP-positive cells were sorted. Knockdown efficiency was confirmed by RT-PCR. The shRNA sequence targeting $ALDH1A1$ is GGAGTGTTTACCAAAGACATT. For knockdown of $C/EBP\beta$, $EP300$, or $NR3C1$, HPNE cells with $ARID1A$ knockout were transfected with plasmid pGIPz-$EP300$ or pGIPz-$NR3C1$. After infection for 5 days, the GFP-positive cells were sorted. Knockdown efficiency was confirmed by RT-PCR. The shRNA sequences targeting $NR3C1$ are CCAACGGTGGCAATGTGAA and AGCTGTAAAGTTTTCTTCA; and targeting $EP300$ are CGGCAAA-CAGTTGTGCACA and AGCTACTGAAGATAGATTA.

## Primary acinar isolation and culture

Pancreata isolated from KC mice and AKC mice 1 month after 5-day tamoxifen administration were rinsed twice in cold 1× HBSS buffer, then minced into small pieces and digested with digestion buffer (HBSS buffer with 10 mM HEPES and 0.5 mg/ml collagenase and 0.25 mg/ml trypsin inhibitor) for 20–30 min at 37°C. During the incubation. the tissue was pipetted every 5 min. After washing twice with washing buffer (HBSS buffer with 5% FBS), the digested tissue was resuspended with media (Waymouth media with 2.5% FBS and 0.25 mg/ml trypsin inhibitor and 100 U/ml Penicillin-Streptomycin), filtrated with 100 μm strainer and seeded into 10 cm dishes overnight at 37°C to remove fibroblasts and ductal cells. The unattached acinar cells were then transferred into collagen-coated plates for growth. To activate Kras expression, 25 ng/ml EGF was added into the media for 5 days. Cells were then used for SA-β-Gal staining, RT-PCR, and western blot.

## Western blot

Western blot was performed using the standard protocol. Antibodies used in this study include ALDH1A1 (Abcam Cat# ab23375, RRID:AB_2224009), ALDH3A1 (Abcam Cat# ab76976, RRID:AB_1523110), KRAS (Abcam Cat# ab180772, RRID:AB_2884935), phosph-Erk1/2 (Cell Signaling Technology Cat# 4370, RRID:AB_2315112), and β-actin (Sigma-Aldrich Cat# A1978, RRID:AB_476692).

## Colony formation assay

HPNE cells ($3 \times 10^4$/well) were seeded into 6-well plates. The cells were treated with doxycycline (6 μg/ml for 15 days) with and without DEAB (1.5 μM for 30 days, Stemcell Technologies Inc 01705).

The media was changed every 2 days. When the colonies were large enough, the cells were fixed and stained with crystal violet.

## ROS measurement

HPNE cells were treated with doxycycline (6 μg/ml for 5 days). ROS level was measured by Flow Cytometry using ROS Detection Assay Kit (BioVision, Cat# K936-250).

## SA-β-Gal staining

SA-β-Gal staining was performed on slides of freshly frozen tissues or cells using Senescence β-Galactosidase Staining Kit (Cell Signaling Technology, Cat# 9860). Total and SA-β-Gal-positive lesions or cells were counted at random fields under the microscope, and positive rates were calculated. For quantification of SA-β-Gal staining of primary acinar cells, due to the difficulty of recognizing the nuclei, average optical density (OD) was used to quantify the intensity of SA-β-Gal staining. 8-bit images were adjusted for white balance and color-deconvoluted using Feulgen light green vector in ImageJ. The average gray values of the green channel were measured. OD was calculated using the following formula: $OD = \log_{10}(255/\text{gray value})$.

## PanIN quantification

Tissues were fixed in 4% paraformaldehyde overnight, processed, and embedded in paraffin. Paraffin-embedded sections were subjected to hematoxylin and eosin staining (H&E staining). ADM, mPanIN-1A, mPanIN-1B, mPanIN-2, and mPanIN-3 were quantified using ImageJ for morphometric analysis based on scanned H&E slides at 20× magnification. Grades of lesions were determined based on the characteristic morphology criterion (*Gopinathan et al., 2015*) with consulting of pathology core. PanIN-1A: flat epithelium composed of columnar cells with basally oriented nuclei. PanIN-1B: identical to PanIN-1A lesions but exhibit papillary or basally pseudostratified architecture. PanIN-2: show mild nuclear abnormalities, including loss of polarity, nuclear enlargement, nuclear crowding, and nuclear pleomorphism. PanIN-3: show a predominantly papillary or micropapillary architecture with abnormal cribriforming, budding, and luminal necrosis; more severe cytological atypia, such as loss of nuclear polarity, dystrophic goblet cells, nuclear irregularities, and macro nucleoli. In case the lesion harbors more than one PanIN grade, the lesion was graded based on the component with the highest grade. Numbers of lesions of different grades were counted for at least five fields of view. The area of tissue was measured for each field of view. Lymph nodes of the pancreatic area were excluded. Numbers of lesions and tissue areas were summed up to calculate lesion number per area.

## IHC quantification

For quantification of IHC results against ALDH3A1, H-score method was used. In brief, staining intensity (not stained: 0; weakly stained: +1; moderately stained: +2; or strongly stained: + 3) was determined for each lesion of interest in the field. The H-score was calculated by the following formula: 3 × percentage of strongly stained cells + 2 × percentage of moderately stained cells + 1 × weakly stained cells, giving a range of 0–300.

## Bulk RNA-seq

HPNE cells were treated with doxycycline (6 μg/ml) for 5 days. RNA samples were prepared using the standard protocol for Trizol. mRNA was enriched using NEBNext Poly(A) mRNA Magnetic Isolation Module (NEB, E7490), and the library was prepared using the NEBNext Ultra II RNA Library Prep Kit for Illumina (NEB, E7770). All libraries were sequenced on Illumina Nextseq500 platform. Reads were aligned to hg19 assembly of the human genome by STAR aligner (*Dobin et al., 2013*), and transcripts counting was performed by HTseq-count (*Anders et al., 2015*). Differential gene expression analysis was performed by using edgeR (*Robinson et al., 2010*) with a cutoff of FDR at 0.05. To identify the genes with differential response to oncogenic KRAS in KO and WT cells, we also performed the interaction analysis in edgeR.

## Analysis of *ALDH1A1* expression in normal pancreas and PDAC

The expression profiles of ALDH genes in normal pancreas were obtained from GTEx database. The expression level of *ALDH1A1* in different cell types in normal pancreas was obtained from Human

Protein Atlas database. The PDAC RNA-seq data were from ICGC-PACA-AU cohort. The raw count data were downloaded from https://dcc.icgc.org/https://dcc.icgc.org/https://dcc.icgc.org/https://dcc.icgc.org/.

## ATAC-seq experiment

ATAC-seq was performed following the protocol of Howard Chang's lab (https://www.nature.com/articles/nmeth.4396) with slight modifications. In brief, $5 \times 10^4$ cells were lysed with ATAC-Resuspension Buffer (RSB) containing 0.1% NP40 and 0.1% Tween-20. After incubation on ice for 3 min, the cell lysates were washed by RSB with 0.1% Tween-20. The cell lysates were then incubated with transposition mixture at 37°C for 30 min. After amplification, the transposed fragments were purified with magnetic beads. Finally, 4 ng fragments were used for the generation of the library. All libraries were sequenced on Illumina Nextseq500 platform.

## ATAC-seq data analysis

Reads were then mapped to the hg19 assembly by Bowtie2 (*Langmead and Salzberg, 2012*) after removing the adaptor sequence. The quality control of ATAC-seq data was performed by using the ATACseqQC R package (*Ou et al., 2018*). Next, the mapped reads from three technical replicates of each genotype were combined for the peak calling by MACS2 (*Zhang et al., 2008*). Peaks from wildtype samples and *ARID1A-KO* samples were combined to get a union peak set. All the peaks were then annotated by HOMER (*Heinz et al., 2010*). HTseq-count (*Anders et al., 2015*) was used for read counting. edgeR package (*Robinson et al., 2010*) was then used for normalization between different samples and for peak differential analysis. The read density profiles of the differential peaks were plotted by deepTools (*Ramírez et al., 2016*). The motif enrichment analysis was performed by using AME.

## ENCODE data analysis

The expression levels of *ALDH1A1* in the cell lines in ENCODE were obtained from the Cancer Cell Line Encyclopedia (CCLE) database (file name: file-CCLE_RNAseq_genes_rpkm_20180929.gct.gz; *Ghandi et al., 2019*). Seven cell lines with RPKM >10 were categorized as ALDH1A1[high] cell lines. Five cell lines with RPKM <0.5 were selected as ALDH1A1[low] cell lines. The A549 H3K27ac and H3K4me1 ChIP-seq data were reanalyzed by using hg19 assembly as described above. The peak files of H3K27ac/H3K4me1 from the other cell lines were directly obtained from the ENCODE database. The ALDH1A1-related differential ATAC peaks, which are overlapped with the H3K27ac/H3K4me1 peaks in at least four out of seven ALDH1A1[high] cell lines, were characterized as functional enhancer regions.

## PanIN-seq

We used the MMI CellCut platform to perform LCM. 40–50% laser power was used with the cutting speed of 18 μm/s to dissect microscopic lesions. 1.6 μl of MATQ-seq lysis buffer (1 μl of 0.2% Triton X100 [Sigma-Aldrich], 0.4 μl of primer mix, 0.12 μl dNTP, 0.05 μl 0.1 M DTT [Life Technologies], and 2 U RNaseOUT [Life Technologies]) was added onto the isolation cap where the dissected tissue was attached (MMI, Prod. No. 50206). We used a pipette tip to scrape the laser-dissected tissue into the lysis buffer and then pipetted the lysis buffer into the tube. Sample tubes were then placed on a thermocycler and incubated at 72°C for 3.5 min, followed by 1 min incubation on ice. 2.4 μl of MATQ-seq first strand synthesis buffer (0.8 μl 5× First Strand Buffer [Life Technologies], 0.2 μl 0.1 M DTT, 4 U RNaseOUT [Life Technologies], 30 U Superscript III [Life Technologies], and 1.15 μl RNase-free water) was then added. The reverse transcription program was same as MATQ-seq. After reverse transcription, the residual primers were then digested by using T4 polymerase at 37° for 40 min and 75° for 20 min. RNA was then digested by using RNase-H and RNase-If at 37° for 15 min and 72° for 15 min. Following that, dC-tailing and second strand synthesis were performed as described in MATQ-seq. The library prep for PanIN-seq samples were same as MATQ-seq. All libraries were sequenced on Illumina Nextseq500 platform.

### PanIN-seq data analysis

The raw sequencing data trimming and barcode retrieval were performed as previously described (*Sheng et al., 2017*). The reads were mapped to the genome MM10 using STAR with the

following parameters: `--outFilterMismatchNoverLmax 0.05 --outFilterMatchNmin 16 --outFilterScoreMinOverLread 0 --outFilterMatchNminOverLread 0`. We used Gencode annotation release mm10 (GRCm38.p4) for transcript annotation. Unique barcode counting and gene expression-level quantification were performed as previously described with a few modifications: the mapping position of the reads was included as part of the identity of the corresponding barcodes; only reads mapped to the exon region were used for gene expression-level quantification. The genes with APM >2 in at least five samples were retained for differential gene expression analysis, which was performed by using edgeR with a cutoff of FDR at 0.05.

## Functional enrichment analysis

Functional enrichment analysis was performed by using GSEA with the MSigDB Hallmark gene sets, and the senescence-related gene sets from the MSigDB curated gene sets. For PanIN-seq, the permutation of phenotype label was used to calculate the p-value.

## Acknowledgements

We are grateful to the McNair family for their support. CZ was also supported by the NIH Director's New Innovator Award (1DP2EB020399). We thank Dr. Zhong Wang (University of Michigan) for providing the Arid1a floxed mice, Dr. Jennifer Bailey (UT Health Center) for providing the HPNE cell line, and Dr. Haoqiang Ying (MD Anderson Cancer Center) for providing inducible KRAS plasmids. We thank Kuanwei Sheng, Ejune Chen, Jinxiang Yuan, and other Zong lab members for their help in this project. We thank Dr. Christophe Herman for his proofreading and helpful discussion.

## Additional information

### Funding

| Funder | Grant reference number | Author |
| --- | --- | --- |
| NIH Office of the Director | 1DP2EB020399 | Chenghang Zong |
| Robert and Janice McNair Foundation | McNair Scholarship | Chenghang Zong |

The funders had no role in study design, data collection and interpretation, or the decision to submit the work for publication.

### Author contributions

Shou Liu, Data curation, Formal analysis, Investigation, Methodology, validation, visualization, Writing – original draft, Writing – review and editing; Wenjian Cao, Yichi Niu, Data curation, Formal analysis, Investigation, Methodology, Writing – original draft, Writing – review and editing; Jiayi Luo, Investigation; Yanhua Zhao, Zhiying Hu, Project administration; Chenghang Zong, Conceptualization, Funding acquisition, Investigation, Methodology, Project administration, Supervision, Writing – original draft, Writing – review and editing

### Author ORCIDs

Yichi Niu http://orcid.org/0000-0002-4376-7792
Jiayi Luo http://orcid.org/0000-0002-9873-0671
Chenghang Zong http://orcid.org/0000-0002-8337-8038

### Ethics

This study was performed in strict accordance with the recommendations in the Guide for the Care and Use of Laboratory Animals of the National Institutes of Health. All of the animals were handled according to approved institutional animal care and use committee (IACUC) protocols (#AN-6434) of Baylor College Medicine. Every effort was made to minimize suffering.

### Decision letter and Author response

Decision letter https://doi.org/10.7554/eLife.64204.sa1
Author response https://doi.org/10.7554/eLife.64204.sa2

## Additional files

### Supplementary files
- Supplementary file 1. Gene set enrichment analysis on PanIN-seq.
- Supplementary file 2. Differential gene expression analysis on PanIN-seq.
- Supplementary file 3. Differential gene expression analysis on RNA-seq.
- Supplementary file 4. Interaction test on the RNA-seq data.
- Supplementary file 5. Quality control of ATAC-seq data.
- Supplementary file 6. ATAC peaks in the regulatory regions of *ALDH1A1*.
- Supplementary file 7. Motif enrichment analysis on differential ATAC peaks.
- Supplementary file 8. Expression (RPKM) of *ALDH1A1* in other human cell lines.
- Supplementary file 9. Binding of TFs at the regulatory region of *ALDH1A1* in human cell lines.
- Supplementary file 10. List of ENCODE datasets used.
- Transparent reporting form

### Data availability
Sequencing data have been deposited in GEO under GSE160444.

The following dataset was generated:

| Author(s) | Year | Dataset title | Dataset URL | Database and Identifier |
|---|---|---|---|---|
| Liu S, Cao WJ, Niu YC, Luo JY, Zhao JH, Zong C, Hu Z | 2020 | | https://www.ncbi.nlm.nih.gov/geo/query/acc.cgi?acc=GSE160444 | NCBI Gene Expression Omnibus, GSE160444 |

The following previously published datasets were used:

| Author(s) | Year | Dataset title | Dataset URL | Database and Identifier |
|---|---|---|---|---|
| The International Cancer Genome Consortium | 2016 | | https://www.ebi.ac.uk/ega/studies/EGAS00001000154 | European Genome-phenome Archive (EGA), EGAS00001000154 |
| The Cancer Genome Atlas Research Network | 2017 | | https://portal.gdc.cancer.gov/projects/TCGA-PAAD | GDC Data Portal, TCGA-PAAD |

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
