## [Decision Letter]

**Acceptance summary:**

The authors have identified the importance of ALDH1A1 in preventing oncogene-induced senescence in precursor lesions of pancreatic cancer. Using a combination of sequencing, computational analysis and experimental work they find that ALDH1A is negatively regulated by ARID1A and that loss of ARID1A results in its upregulation and a decrease in intracellular ROS. This highlights a mechanism whereby PanINs prevent senescence and drive a pro-growth phenotype.

**Decision letter after peer review:**

Thank you for submitting your article "Single-PanIN-seq Unveils that ARID1A Deficiency Promotes Pancreatic Tumorigenesis by Attenuating KRAS Induced Senescence" for consideration by *eLife*. Your article has been reviewed by 3 peer reviewers, and the evaluation has been overseen by Maureen Murphy as the Senior and Reviewing Editor. The reviewers have opted to remain anonymous.

The reviewers have discussed the reviews with one another and the Reviewing Editor has drafted this decision to help you prepare a revised submission. Please be advised that there was considerable discussion about your work, and that considerable revisions are requested by the reviewers. You may wish to create a plan of attack for the revisions and send this to the Senior Editor, who can share it with reviewers.

Summary:

The authors have identified the importance of ALDH1A1 in preventing oncogene-induced senescence in precursor lesions of pancreatic cancer. By a combination of sequencing, computational analysis and experimental work they find that ALDH1A is negatively regulated by ARID1A and that loss of ARID1A results in its upregulation and a decrease in intracellular ROS. This highlights another new mechanism that is exploited by PanINs to prevent senescence and drive a pro-growth phenotype.

Essential revisions:

1. The conclusion that "a highly complex trans-differentiation occurs following the knockout of Arid1a" is not supported by the evidence provided in Figure S3. It is unclear what panels B and C are referring to (no labels) and there is no scale given. The authors also use the RNA-sequencing data to conclude that "the activities of Kras signaling are partially suppressed by ARID1A deficiency." This is a major point and should be supported by experimental evidence in their cell models.

2. Although Figure S5B shows that genetic manipulation of ARID1A was successful, there is no evidence by western blot or PCR that ARID1A has truly been knocked out in their HPNE cells. This model was used for a large portion of the data and must be characterized in a convincing way.

3. The evidence for senescence in vitro is not strong. Several times throughout, a colony formation assay is used as a measure of senescence, rather than SA-b-gal, molecular markers or secreted factors. It is unclear why the authors measured SA-b-gal staining intensity rather than %-positive cells. In particular, the evidence that ARD1A regulates Kras-induced senescence in HPNE cells is lacking.

4. It is unclear if the ROS phenotype is mediated by ALDH1A1. Therefore, the conclusion that "ARID1A knockout can effectively reduce cellular ROS level by upregulating the expression of ALDH1A1" is not supported.

5. There is no functional experimental evidence provided for how ARID1A regulates the expression of ALDH1A1. This is a major point of novelty in the paper and should therefore be supported by experimental evidence.

6. In the patients with pancreatic ductal adenocarcinoma, typically one sees point mutations, InDel or structure variations in the cancer genes; can the authors list the mutations in ARID1A in their samples to see if there are any recurrent mutations (i.e, visualized by lollipop plot, using ICGC-PACA-AU or ICGC-PACA-CA whole genome sequencing dataset, https://dcc.icgc.org/)? Can the authors perform an eQTL analysis (for example using FastQTL) to see if any mutations in ARID1A and +/- 1Mb cis-mutations are associated with the expression of ALDH1A1? The mutations and gene expression datasets could be readily downloaded from ICGC-PACA-AU or ICGC-PACA-CA (ICGC data portal).

7. The authors describe differences in proportions of distinct populations of ADM, PanIN-1, PanIN-2, PanIN-3 populations. At no point in their introduction/results do they describe the PanIN-1/2/3 populations. This would be a good addition for clarity. In addition there are no details on how these were quantified in the methods. In the introduction the authors state that previously ARID1A KO has been found to increased ADM – however Figure S1 shows the opposite that ARID1A KO leads to decreased ADM – while this reflects a trajectory of PanIN progression – not including any background on this is confusing and needs clarification.

8. How many genes in total were differentially expressed following MATQ-seq? Were lowly expressed genes removed? Was there any requirement that a gene was found to be expressed in at least N samples etc? These kind of filters are standard for scRNA-seq analysis and would seem to be appropriate for this type of data.

9. Using multiple pairwise comparisons is a restrictive way of analyzing this data – the authors should consider re-analyzing this data using an interaction test – trying to identify genes that respond differently to KRAS induction dependent on whether ARID1A is knocked out or not. Or consider an approach based on clustering of expression across differentially expressed genes to identify patterns of response across all four conditions. One prediction of the model the authors are proposing should be that SASP genes are upregulated in the KRAS vs WT but blunted in KRAS-ARID1A vs WT – is this observed?

10. Figure 3F: Figure 3F – please add a title containing the gene name – having the y-axis as ABM is confusing at first glance. Any supplementary table/data to show that ALDH1A1 was not in the list of DEG in AKC mice? Or to show that ALDH3A1 was? Is there a difference in regulation between species (i.e. both ALDH1A1 and ALDH3A1 are expressed, but only ALDH1A1 responds) or that only ALDH1A1 is expressed?

11. There is no comprehensive analysis linking the results from the in vitro MATQ-seq in mouse and the in vitro work in human. There is no discussion of whether orthologous genes are seen to up/downregulated in both. Linking together the results from the AKC vs KC lesions with KRAS-ARID1A-KO vs KRAS-Wildtype HPNE cells would allow the results from both models to be compared – which would strengthen the results presented.

12. Figure 4A: What is the expression of ALDH family members in the normal pancreas? This is really needed to make a comparison and statement as presented by the authors. In addition, Figure 4A is really showing high expression of ALDH1A1 and ALDH2 and not all "ALDH family proteins".

13. Figure 4H: The authors have demonstrated no evidence that in PanINs KRAS is a target of ARID1A? Was this observed in their analysis of HPFE cells? The origin of this link is not apparent.

14. ATAC-seq analysis: there is no analysis of the gained or lost regions at all. It is not stated in the text as to the numbers of gained/lost nor is there any analysis of the genes they are near (i.e. using GREAT or something similar). Are the sites that have significant changes in accessibility enriched for specific TFBSs etc.… i.e. other papers have suggested that ARID1A is a co-factor for AP-1 etc.…. (https://clinicalepigeneticsjournal.biomedcentral.com/articles/10.1186/s13148-019-0690-5).

15. Figure 5: Figure 5A: the text uses the phrase that Spearmans correlation coefficients are significantly lower.….. this is implying a hypothesis test – please rephrase. Figure 5B/C: needs to be clarified on how it shows "changes in accessibility of regulatory regions were significantly correlated with the alterations in gene expression levels in ARID1A-KO cells". How many of the 311 gained promoter ATAC regions overlap with genes which are upregulated (and vice versa)?

16. Supp. Table S4: There are 12 peaks reported in this table of which 10 are significantly "gained" this is different to the numbers reported in the manuscript. Please clarify/correct.

17. Figure S10: What is significance of identifying TFs? What are the significance of these TFs? Have they appeared in previous publications? Are any of these changing expression in ARID1A-KO RNA-seq? This is an underdeveloped analysis of what looks like good and interesting data.

18. Not all of the results from the differential expression analyses are available as Supplemental tables – this should be fixed.

19. In several parts of the manuscript – significance testing was carried out on only two data points (there can be no reliable estimate of variance using only two data points) using parametric methods – this is likely giving a false impression of significance – please either increase N or use a more appropriate test.

---

## [Author Response]

Essential revisions:1. The conclusion that "a highly complex trans-differentiation occurs following the knockout of Arid1a" is not supported by the evidence provided in Figure S3. It is unclear what panels B and C are referring to (no labels) and there is no scale given.

After more careful thinking, we agree with the reviewer’s critique. It is difficult to prove the transdifferentiation processes in vivo. In the revision, we deleted the related figure and descriptions.

The authors also use the RNA-sequencing data to conclude that "the activities of Kras signaling are partially suppressed by ARID1A deficiency." This is a major point and should be supported by experimental evidence in their cell models.

To provide the experimental support as suggested, we performed the western blot of ERK on ARID1A-KO and wild type HPNE cells and found that the activation of ERK signaling by KRAS was impaired in ARID1A-KO HPNE cells. We have added the new result to Figure 3—figure supplement 2.

2. Although Figure S5B shows that genetic manipulation of ARID1A was successful, there is no evidence by western blot or PCR that ARID1A has truly been knocked out in their HPNE cells. This model was used for a large portion of the data and must be characterized in a convincing way.

Following the reviewer’s suggestion, we have provided the qRT-PCR result (Figure 2—figure supplement 1C) to demonstrate that the expression of ARID1A is indeed significantly reduced in our HPNE cells with ARID1A knockout.

3. The evidence for senescence in vitro is not strong. Several times throughout, a colony formation assay is used as a measure of senescence, rather than SA-b-gal, molecular markers or secreted factors. It is unclear why the authors measured SA-b-gal staining intensity rather than %-positive cells.

First, we would like to point out that we have performed SA-b-gal staining on in vitro cells. However, we observed that the difference between ARID1A-KO and WT cells was not as significant as in vivo (Figure 2—figure supplement 1D). We have included this staining result in our first version. We attributed this result to the potential inefficiency in using SA-b-gal staining to identify the senescent HPNE.

The reason that we chose to measure SA-b-gal staining intensity in ex vivo experiment (Figure 2C-D) is as following: we found out that it was difficult to count the SA-b-gal negative cells due to the flat morphology of senescent cells and the large cell-size variations. Without accurate counting of negative cells, we cannot achieve the accurate calculation of the percentage of positive cells. We have added additional explanation in the revised manuscript to clarify the reason. To further clarify, for in vivo measurement, we still use %-positive analysis.

In particular, the evidence that ARD1A regulates Kras-induced senescence in HPNE cells is lacking.

To address the criticism for lack of evidence, we added the following experimental data in the revised manuscript: we examined the expression levels of *CDKN1A* in HPNE cell line model since *CDKN1A* is classical marker of cellular senescence. In the RNA-seq data, we observed that the expression of *CDKN1A* upon KRAS induction is also significantly suppressed in ARID1A-KO cells in comparison to wild type cells (Figure 3—figure supplement 3C). This result was further verified by qRT-PCR, as shown in Figure 3—figure supplement 3D. This new result provides additional experimental evidence supporting that ARID1A regulates KRAS-induced senescence in HPNE cells.

4. It is unclear if the ROS phenotype is mediated by ALDH1A1. Therefore, the conclusion that "ARID1A knockout can effectively reduce cellular ROS level by upregulating the expression of ALDH1A1" is not supported.

To provide the direct support for the role of ALDH1A1 in reducing cellular ROS, we added the ALDH1A1 knockdown experiment in the revision. The knockdown efficiency is shown by qRTPCR in Figure 4—figure supplement 2. We measured Kras-induced ROS level in ARID1A knockout HPNE cells with or without ALDH1A1 knockdown. As a result, ROS level in ALDH1A1 knockdown cells was significantly higher than in the cells with scramble shRNA upon Kras induction, indicating that ALDH1A1 is responsible for the clearance of ROS in HPNE cells with *ARID1A* knockout. This new result has been added to Figure 4H.

5. There is no functional experimental evidence provided for how ARID1A regulates the expression of ALDH1A1. This is a major point of novelty in the paper and should therefore be supported by experimental evidence.

In our first version of manuscript, we showed that *ARID1A* knockout could significantly increase the chromatin accessibility at the enhancer regions of *ALDH1A1* (Figure 6A) in our ATAC-seq analysis. By analyzing the published ChIP-seq datasets of different TFs, we have identified the candidate factors that are involved in the regulation of *ALDH1A1* expression. In particular, EP300 and NR3C1 could bind to two enhancer loci of *ALDH1A1* (Figure 6C, left panel).

Here, to verify the predicted TFs based on ATAC-seq data as well as address this criticism for lack of experimental evidence on the role of ARID1A in regulating the expression of ALDH1A1, we have knocked down *EP300* and *NR3C1*, respectively, in *ARID1A*-KO HPNE cells (Figure 6figure supplement 1). As shown in Figure 6C (middle and right panels), knockdown of *EP300* or *NR3C1* significantly impairs the transcription of *ALDH1A1* gene. As a control, we also showed that the expression of EP300 and NR3C1 are not changed between ARID1A-KO cells and wild type cells (Supplementary File 3). Therefore, these new experimental data support the following regulatory mechanism: ARID1A deficiency promotes the accessibility of the enhancer regions of ALDH1A1, which leads to the binding of EP300 and NR3C1 to the ALDH1A1 enhancers and the activation of ALDH1A1 expression.

6. In the patients with pancreatic ductal adenocarcinoma, typically one sees point mutations, InDel or structure variations in the cancer genes; can the authors list the mutations in ARID1A in their samples to see if there are any recurrent mutations (i.e, visualized by lollipop plot, using ICGC-PACA-AU or ICGC-PACA-CA whole genome sequencing dataset, https://dcc.icgc.org/)?

To address this question, we have analyzed the mutation data from two cohorts (ICGC-PACA-AU and TCGA) and plotted the ‘Lollipop’ plots by using the MutationMapper web tool at http://www.cbioportal.org/. First, we didn’t find the recurrent mutations in the coding regions of ARID1A in these PDAC patients (Figure 1—figure supplement 1). It is worth noting that most of these mutations generate truncated proteins, which further support the roles of loss-of-function mutations in ARID1A.

Can the authors perform an eQTL analysis (for example using FastQTL) to see if any mutations in ARID1A and +/- 1Mb cis-mutations are associated with the expression of ALDH1A1? The mutations and gene expression datasets could be readily downloaded from ICGC-PACA-AU or ICGC-PACA-CA (ICGC data portal).

To test the eQTL based analysis, we have summarized the somatic mutations identified in ICGCPACA-AU and ICGC-PACA-CA datasets. In total, we identified 1510 somatic SNVs within +/- 1Mb regions of ARID1A. However, only 16 of them are recurrent mutations detected in multiple donors (15 mutations detected in two donors and 1 mutation detected in three donors). In the eQTL analysis, the SNVs in query need to be detected in multiple donors to reduce person-to-person variation and warrant statistical power. Here, considering the low number of recurrent mutations based on the two PDAC somatic SNV datasets, we conclude that we do not have statistical power to identify the mutations that are associated with the expression of ALDH1A1. Furthermore, we also examined whether there is significant enrichment of variants around the potential enhancer elements of ARID1A gene based on ATAC-seq data. However, we did not find any significant clustering of variants, which prevents us from performing meaningful eQTL analysis.

7. The authors describe differences in proportions of distinct populations of ADM, PanIN-1, PanIN-2, PanIN-3 populations. At no point in their introduction/results do they describe the PanIN-1/2/3 populations. This would be a good addition for clarity. In addition there are no details on how these were quantified in the methods.

As suggested, we added the details on lesion quantification in the method section in Supplementary Materials (Section PanIN quantification, Page 4).

In the introduction the authors state that previously ARID1A KO has been found to increased ADM – however Figure S1 shows the opposite that ARID1A KO leads to decreased ADM – while this reflects a trajectory of PanIN progression – not including any background on this is confusing and needs clarification.

We agree with the reviewer’s assessment. We have added additional description in the Main text (last paragraph, Page 4). Here, we would like to emphasize that in the previous study by Livshits *et al.* (Reference 8), the authors observed increase of ADM in adult mice two weeks after *Arid1a* was knocked out. In our study, we examined the effects of *Arid1a* knockout on PanIN progression at much later stage, from 2 months to 6 months after *Arid1a* knockout. Indeed, as the reviewer pointed out, this reflects a trajectory of PanIN progression.

8. How many genes in total were differentially expressed following MATQ-seq? Were lowly expressed genes removed? Was there any requirement that a gene was found to be expressed in at least N samples etc? These kind of filters are standard for scRNA-seq analysis and would seem to be appropriate for this type of data.

To address this critique, we have added the number of the differentially expressed genes into our manuscript (Supplementary File S2). The reviewer is also right. As a standard procedure, we do have a hard cutoff to remove lowly expressed genes when we performed DEG analysis on MATQseq. We only retained the genes with APM > 2 in at least five samples for DEG analysis. We apologize for missing some of the analysis information in our first version and we have added the details in the revised manuscript.

9. Using multiple pairwise comparisons is a restrictive way of analyzing this data – the authors should consider re-analyzing this data using an interaction test – trying to identify genes that respond differently to KRAS induction dependent on whether ARID1A is knocked out or not. Or consider an approach based on clustering of expression across differentially expressed genes to identify patterns of response across all four conditions. One prediction of the model the authors are proposing should be that SASP genes are upregulated in the KRAS vs WT but blunted in KRAS-ARID1A vs WT – is this observed?

We thank the reviewer for this suggestion. We have performed an interaction test by using edgeR to identify the genes that respond differentially to KRAS induction depending on the mutation status of ARID1A and we have added the results in the revised manuscript. To summarize, with the interaction test, we identified 156 genes whose responses to KRAS activation are significantly different between ARID1A-KO and wild type HPNE cells (FDR < 0.05) (Supplementary File S4). Next, we ranked the genes based on their differences in the response to KRAS induction (measured by the fold change in the output of edgeR DEG test) between *ARID1A*-KO and WT cells. Gene set enrichment analysis (GSEA) was then performed on the ranked gene list (Figure 3—figure supplement 3A-B). As a result, we observed that the activation of 4 inflammation-related pathways upon Kras induction (including interferon-α response, interferon-γ response, inflammatory response and TNF-α signaling via NF-κB pathways) are significantly suppressed in *ARID1A*-KO HPNE cells compared to WT cells, which suggests that these pathways are blunted in *ARID1A*-KO cells upon KRAS induction.

10. Figure 3F: Figure 3F – please add a title containing the gene name – having the y-axis as ABM is confusing at first glance.

To address this comment, we have added the gene name to Figure 3F. We have also modified the y-axis label as “gene expression levels (APM)” and the definition of the unit APM is given in the caption of the figure.

Any supplementary table/data to show that ALDH1A1 was not in the list of DEG in AKC mice? Or to show that ALDH3A1 was?

Yes, we included the full list of DEG table of our PanIN-seq data in Supplementary File S2 to show that Aldh3a1, but not Aldh1a1, is significantly upregulated in AKC mice.

Is there a difference in regulation between species (i.e. both ALDH1A1 and ALDH3A1 are expressed, but only ALDH1A1 responds) or that only ALDH1A1 is expressed?

Yes, there are species-based differences. ALDH3A1 is not expressed in HPNE cell line, but it is expressed in mouse model, vice versa for ALDH1A1 in HPNE cell line. Therefore, we do not have a scenario that both ALDH1A1 and ALDH3A1 are expressed, but only ALDH1A1 responds. But the species difference still indicates the regulation could be different. In our regulation analysis with ATAC-seq data and experimental verification, we mainly focus on the regulation of ALDH1A1 for human samples.

11. There is no comprehensive analysis linking the results from the in vitro MATQ-seq in mouse and the in vitro work in human. There is no discussion of whether orthologous genes are seen to up/downregulated in both. Linking together the results from the AKC vs KC lesions with KRAS-ARID1A-KO vs KRAS-Wildtype HPNE cells would allow the results from both models to be compared – which would strengthen the results presented.

To address this critique, we have compared the results from the in vivo mouse model to the in vitro human cell line model. We only find a limited number of genes that are significantly downregulated or upregulated in both (10 genes, the expected number of shared genes is 5.3). We speculate that the reasons for the limited number of orthologous genes sharing similar changes include cell type difference (the cell that we sequenced by PanIN-seq exist in the intermediate state of cell identity changes, while the HPNE has the fully established identity as epithelia cells) and the species difference.

Despite the limited number of orthologous genes sharing similar changes, we found that the perturbed pathways are quite similar between these two models. Among the 11 pathways significantly downregulated (FDR < 0.05) in *ARID1A*-KO cells under KRAS induction, 8 of them are also significantly downregulated (FDR < 0.05) in AKC lesions compared to KC lesions (the expected number is 3.74) (the shared pathways are labeled with asterisk in Figure 3—figure supplement 1). Here we didn’t compare the upregulated pathways in both models since the number of pathways that are upregulated in AKC lesions is small (only two pathways with FDR <0.1).

Overall, this new analysis suggests that while there are gene expression differences between these two model systems (mouse versus human, in vivo versus in vitro*,* and the neoplasia versus epithelia like cells), however, the general effects of ARID1A deficiency on the lesion cells are consistent across species as well as between in vivo and in vitro systems. We have clarified these points in the revised manuscript and added the corresponding analysis in Figure 3—figure supplement 1.

12. Figure 4A: What is the expression of ALDH family members in the normal pancreas? This is really needed to make a comparison and statement as presented by the authors.

To address this question, we have investigated the expression of ALDH family members in the normal pancreas and found that ALDH1A1 is the main subtype expressed in the normal pancreas (Figure 4—figure supplement 1A). With the published single cell RNA-seq data, we also observed that the expression of *ALDH1A1* varies across different cell types, and the endocrine cells have the highest expression level, followed by exocrine glandular cells (Figure 4—figure supplement 1B).

As suggested by the reviewers, we also compared the expression of *ALDH1A1* in PDAC to normal pancreas. Since the tumor cells are mainly epithelial cells, we only compared PDAC data to pancreatic ductal cells to avoid the confounding factors caused by the cell type difference. As shown in Figure 4—figure supplement 1B, there are four subclusters of ductal cells. The average expression level of *ALDH1A1* in normal pancreatic ductal cells (cluster 1,2 and 3) is less than 50. we exclude the cluster 4 since ALDH1A1 positive cells in ductal cells correspond to the stem cell population. In PDAC patient cohort, we observed that 63% of samples have the expression level of *ALDH1A1* larger than 50 TPM and 10% of samples have ALDH1A1 expression higher than 200 TPM (Figure 4—figure supplement 1C). These observations strongly suggest that ALDH1A1 plays important roles in a significant portion of PDAC tumors.

In addition, Figure 4A is really showing high expression of ALDH1A1 and ALDH2 and not all "ALDH family proteins".

We have modified the main text based on the reviewers’ suggestions to make the description more accurate.

13. Figure 4H: The authors have demonstrated no evidence that in PanINs KRAS is a target of ARID1A? Was this observed in their analysis of HPFE cells? The origin of this link is not apparent.

Based on our PanIN-seq data, we concluded that the activities of KRAS signaling pathway are clearly suppressed in AKC lesions compared to KC lesions. In term of studying the regulation, due to the limited number of cells in PanIN lesion, we were not able to perform ATAC-seq for the lesion cells, which limits our conclusion on whether KRAS is a direct target of ARID1A. On the other hand, since we also observed the similar effect on KRAS pathways upon ARID1A knockout in the cell line model, we can use our cell line data to investigate the potential regulatory relationship between ARID1A and KRAS signaling pathway.

To do so, we analyzed the changes of chromatin accessibility the genes in KRAS signaling pathway (the genes in Hallmark_KRAS_Signaling_UP gene set from MSigDB) by using the ATAC-seq of our HPNE cell data. As a result, we found that the fold-changes of chromatin accessibility of the genes involved in KRAS signaling pathway are significantly lower than that of randomly selected genes (Figure 5—figure supplement 5). This result suggests that ARID1A engages in the regulation of the genes involved in KRAS signaling pathway by modifying the chromatin accessibility of their regulatory regions. We have modified the description in our revised manuscript to explain this potential regulation mechanism (Page 13, first paragraph, Main text).

14. ATAC-seq analysis: there is no analysis of the gained or lost regions at all. It is not stated in the text as to the numbers of gained/lost nor is there any analysis of the genes they are near (i.e. using GREAT or something similar).

Since we focused on the regulatory function of ARID1A on the expression of ALDH1A1 in our manuscript, so we did not include the detailed analysis of ATAC-seq data in our first version. To address this critique, we also added the result of the global analysis of our ATAC-seq data using GREAT. Consistent with Figure 5C-D, the peaks with significantly changed accessibility between ARID1A-KO and WT cells are enriched at the distal regions (these new results are added to Figure 5—figure supplement 3A-B). The number of peaks with differential accessibility are all presented in the new figures. We also added the result of the GO enrichment analysis in GREAT in Figure 5—figure supplement 3C-D.

Are the sites that have significant changes in accessibility enriched for specific TFBSs etc.… i.e. other papers have suggested that ARID1A is a co-factor for AP-1 etc.…. (https://clinicalepigeneticsjournal.biomedcentral.com/articles/10.1186/s13148-019-0690-5).

To address this comment, we performed the motif enrichment analysis by using AME function in MEME. Briefly, we separate the peaks with significant changes into four types: distal peaks with increased accessibility, distal peaks with decreased accessibility, promoter peaks with increased accessibility and promoter peaks with decreased accessibility. We found the peaks with increased accessibility (including both promoter peaks and distal peaks) are enriched with FOX family TFs, SRY, CDX1, SOX5, TEAD4 etc. (Figure 5F, Figure 5—figure supplement 5A and Supplementary File S7). On the other hand, the peaks with decreased accessibility are enriched with FOS/JUN TF family, NFE2, NF2L1 etc., (Figure 5G, Figure 5—figure supplement 5B and Supplementary File S7). These results are consistent with previous reports as mentioned by the reviewers. We also added the suggested reference in the revised manuscript.

15. Figure 5: Figure 5A: the text uses the phrase that Spearmans correlation coefficients are significantly lower.….. this is implying a hypothesis test – please rephrase.

As suggested, we have rephrased our description. Furthermore, we provide the statistical test for Spearman’s correlation coefficients in Figure 5—figure supplement 1G.

Figure 5B/C: needs to be clarified on how it shows "changes in accessibility of regulatory regions were significantly correlated with the alterations in gene expression levels in ARID1A-KO cells".

To address this comment, we have added more detailed descriptions about the procedures in the revised manuscript (Page 12, the second paragraph). In details, we separated the DEGs identified in RNA-seq between ARID1A-KO and WT cells into two groups based on the fold-change (Figure 5B). The genes with positive log-fold change values are the genes upregulated in ARID1A-KO cells (Up-regulated genes group in Figure 5B) and the genes with negative log-foldchange values are the ones downregulated in ARID1A-KO cells (down-regulated genes group in Figure 5B). Next, we calculated the fold-change of chromatin accessibility for each of these DEGs and plotted the distribution of fold-changes of chromatin accessibility for these two groups of genes. As a result, we observed that the average fold-change of chromatin accessibility of upregulated genes is significantly higher than that of down-regulated genes (Figure 5B), which supports our statement.

How many of the 311 gained promoter ATAC regions overlap with genes which are upregulated (and vice versa)?

We performed the suggested analysis to examine the association between differentially expressed genes and the peaks with differential accessibility for both promoters and enhancers. As shown in Figure 5—figure supplement 4A-D, we observed that DEGs are significantly associated with the promoter peaks and the distal peaks with significantly differential accessibility. Furthermore, we noticed that the number of differentially expressed genes associated with the enhancer elements is larger than the number of genes associated with the promoters. This result is consistent with the observation that ARID1A knockout alters gene expression mainly by modulating the chromatin accessibility of the enhancers elements. A new paragraph describing this result (Page 12, last paragraph, Main text) is added in the revised manuscript.

16. Supp. Table S4: There are 12 peaks reported in this table of which 10 are significantly "gained" this is different to the numbers reported in the manuscript. Please clarify/correct.

Our description is probably not very clear. In the new Supplementary File S6, we listed all the ATAC-seq peaks that are linked to ALDH1A1 including both distal peaks and promoter peaks. In the Main text, we only described the peaks in the distal regions, therefore there are 11 peaks. We have added one sentence to clarify this point.

17. Figure S10: What is significance of identifying TFs? What are the significance of these TFs? Have they appeared in previous publications? Are any of these changing expression in ARID1A-KO RNA-seq? This is an underdeveloped analysis of what looks like good and interesting data.

Based on the literature search, we did not find previous reports showing that NR3C1 and EP300 are involved in the regulation of ALDH1A1 in cellular senescence. Here, we unveil how ARID1A mediate the gene expression of ALDH1A1 by these two genes for the first time, which shows the significance of identifying these TFs involved in ARID1A regulation.

There are no expression changes for both of the two genes between wildtype and ARID1A KO. To deepen our study, we also knocked down EP300 and NR3C1, respectively, in *ARID1A*-KO HPNE cells (Figure 6—figure supplement 1). As shown in Figure 6C (middle and right panels), knockdown of EP300 or NR3C1 indeed significantly impairs the transcription of ALDH1A1 gene.

18. Not all of the results from the differential expression analyses are available as Supplemental tables – this should be fixed.

In our first version of manuscript, we only included the genes that reach the statistical significance (FDR < 0.05). To address this critique, we have now provided the full table of DEG results in the original manuscript in Supplementary File S2.

19. In several parts of the manuscript – significance testing was carried out on only two data points (there can be no reliable estimate of variance using only two data points) using parametric methods – this is likely giving a false impression of significance – please either increase N or use a more appropriate test.

We apologize for our description that may cause the confusion about our experimental data and the statistics. For the figures that the reviewer mentioned with two data points, including Figure 2F, Figure 4D and Figure 4G, the two data points corresponds to two clones (clone #2 and clone #11), but for each clone, we did perform 3 biological replicates. We have clarified the description on the corresponding statistical tests.